# Ligand binding remodels protein side-chain conformational heterogeneity

Stephanie A Wankowicz[1,2], Saulo H de Oliveira[3], Daniel W Hogan[1], Henry van den Bedem[1,3], James S Fraser[1]*

[1]Department of Bioengineering and Therapeutic Sciences, University of California, San Francisco, San Francisco, United States; [2]Biophysics Graduate Program, University of California San Francisco, San Francisco, United States; [3]Atomwise Inc., San Francisco, United States

**Abstract** While protein conformational heterogeneity plays an important role in many aspects of biological function, including ligand binding, its impact has been difficult to quantify. Macromolecular X-ray diffraction is commonly interpreted with a static structure, but it can provide information on both the anharmonic and harmonic contributions to conformational heterogeneity. Here, through multiconformer modeling of time- and space-averaged electron density, we measure conformational heterogeneity of 743 stringently matched pairs of crystallographic datasets that reflect unbound/apo and ligand-bound/holo states. When comparing the conformational heterogeneity of side chains, we observe that when binding site residues become more rigid upon ligand binding, distant residues tend to become more flexible, especially in non-solvent-exposed regions. Among ligand properties, we observe increased protein flexibility as the number of hydrogen bonds decreases and relative hydrophobicity increases. Across a series of 13 inhibitor-bound structures of CDK2, we find that conformational heterogeneity is correlated with inhibitor features and identify how conformational changes propagate differences in conformational heterogeneity away from the binding site. Collectively, our findings agree with models emerging from nuclear magnetic resonance studies suggesting that residual side-chain entropy can modulate affinity and point to the need to integrate both static conformational changes and conformational heterogeneity in models of ligand binding.

*For correspondence: jfraser@fraserlab.com

## Editor's evaluation

This work attempts to extract information about protein thermodynamics from X-ray crystallography data, which is a challenging problem. The heterogeneous pattern of order parameter changes in response to ligand binding implies that the approach is identifying new information. This work offers insights into ligand binding affinity and specificity mechanisms, suggesting that distal (allosteric) perturbations represent a possible avenue to modulate protein function.

## Introduction

Ligand binding is essential for many protein functions, including enzyme catalysis, receptor activation, and drug response (**Mobley and Dill, 2009**). Ligand binding reshapes the protein conformational ensemble between the ligand-bound (holo) and unbound (apo) states, stabilizing some conformations and destabilizing others (**Boehr et al., 2009**). Despite the dynamic nature of proteins, when comparing structures, often only static conformational changes are considered. However, differences due to ligand binding can range from large, collective movements, such as a loop closure over the binding pocket, to small, local fluctuations of side chains (**Gutteridge and Thornton, 2005**). Differences in binding affinity and specificity are most often attributed to the enthalpic portion of binding

**eLife digest** Proteins are the workhorses of our cells. They are large molecules that 'fold' into specific, often highly complex, three-dimensional configurations. These structures are not static, but rather dynamic and flexible. In other words, proteins can shift between different three-dimensional shapes to perform their tasks within the cell.

To perform their roles, many proteins have to bind to small molecule ligands. Many ligands are drugs, which means that their effectiveness depends on their ability to bind to and impact the proteins involved in the disease they are treating.

When a ligand binds to a protein, it can reshape the protein. For example, certain conformations of the protein, which were difficult for the protein to be in on its own, may become more stable when the ligand binds. Additionally, upon ligand binding, some parts of the protein may move relative to each other. Previous studies have shown that these movements can affect the interaction between ligand and protein. However, these studies only examined a small number of proteins. Therefore, Wankowicz et al. set out to determine, in greater detail, what happens to protein flexibility upon ligand binding.

First, a pipeline was created to model alternative configurations of the protein both with and without ligands attached. These models measured flexibility within protein structures. The models revealed that when ligands bind to proteins, the flexibility of different regions of the protein changes – and does so in a consistent way. Proteins that become more rigid in the region interacting with their ligands become less rigid in other, distant regions, and vice versa. In other words, the rest of the protein is able to compensate for any changes in flexibility caused by ligand binding, which may contribute to how well a ligand binds to a protein.

This study demonstrates the ability of ligands to affect the entire structure of the proteins they bind to, and therefore sheds new light on the role of proteins' innate conformational flexibility during this process. These results will contribute to our understanding of how the ligands and proteins involved in different cellular processes interact with each other – and, potentially, how these interactions can be manipulated.

free energy, including visualized interactions between the receptor and ligand. On the other hand, conformational heterogeneity, especially side-chain fluctuations, can also contribute energetically to the binding affinity by modulating entropy (*Wand and Sharp, 2018*; *Tzeng and Kalodimos, 2012*). While the individual fluctuation of residues is small, they can add up to significantly contribute to the entropic portion of binding free energy. Previous work examining a diverse set of protein complexes calculated that protein conformational entropy can contribute between –2 (favoring) and 4 (disfavoring) kcal/mol to binding free energy (*Caro et al., 2021*; *Caro et al., 2017*). A holistic understanding of the origins of binding would ideally explore both enthalpic and entropic energetic contributions to binding affinity (*Zhou and Gilson, 2009*).

Side-chain conformational heterogeneity, including jumps between and variation within rotameric conformations, measured by nuclear magnetic resonance (NMR) relaxation studies has been linked to entropy (*Caro et al., 2021*; *Frederick et al., 2007*). In principle, complementary information could be accessed by other structural methods. Structural information from X-ray crystallography or cryo-electron microscopy (CryoEM) typically produces a single set of structural coordinates. However, the underlying density maps are created from thousands-to-millions of protein molecules and averaged in both time and space through the crystal lattice or electron microscope particle stack (*Woldeyes et al., 2014*; *Cheng et al., 2015*). When averaged in a single-density map, conformational heterogeneity across these copies can manifest as 'anharmonic disorder,' which can be modeled using multiple alternative conformations, or 'harmonic disorder,' which can be modeled by B-factors/atomic displacement parameters (*Figure 1A*). Molecular dynamics experiments have demonstrated that if alternative conformations are not modeled correctly and consistently, then B-factors take on values that are not representative of the underlying conformational heterogeneity (*Kuzmanic et al., 2014*; *Kuriyan et al., 1986*). Moreover, B-factors incorporate many effects, including the biases and restraints of the refinement programs, modeling errors, crystal lattice defects, and occupancy changes of atoms. Therefore, consistently modeling X-ray structures as multiconformer models, with alternative side-chain and backbone conformations, along with B-factors, may better complement the view emerging

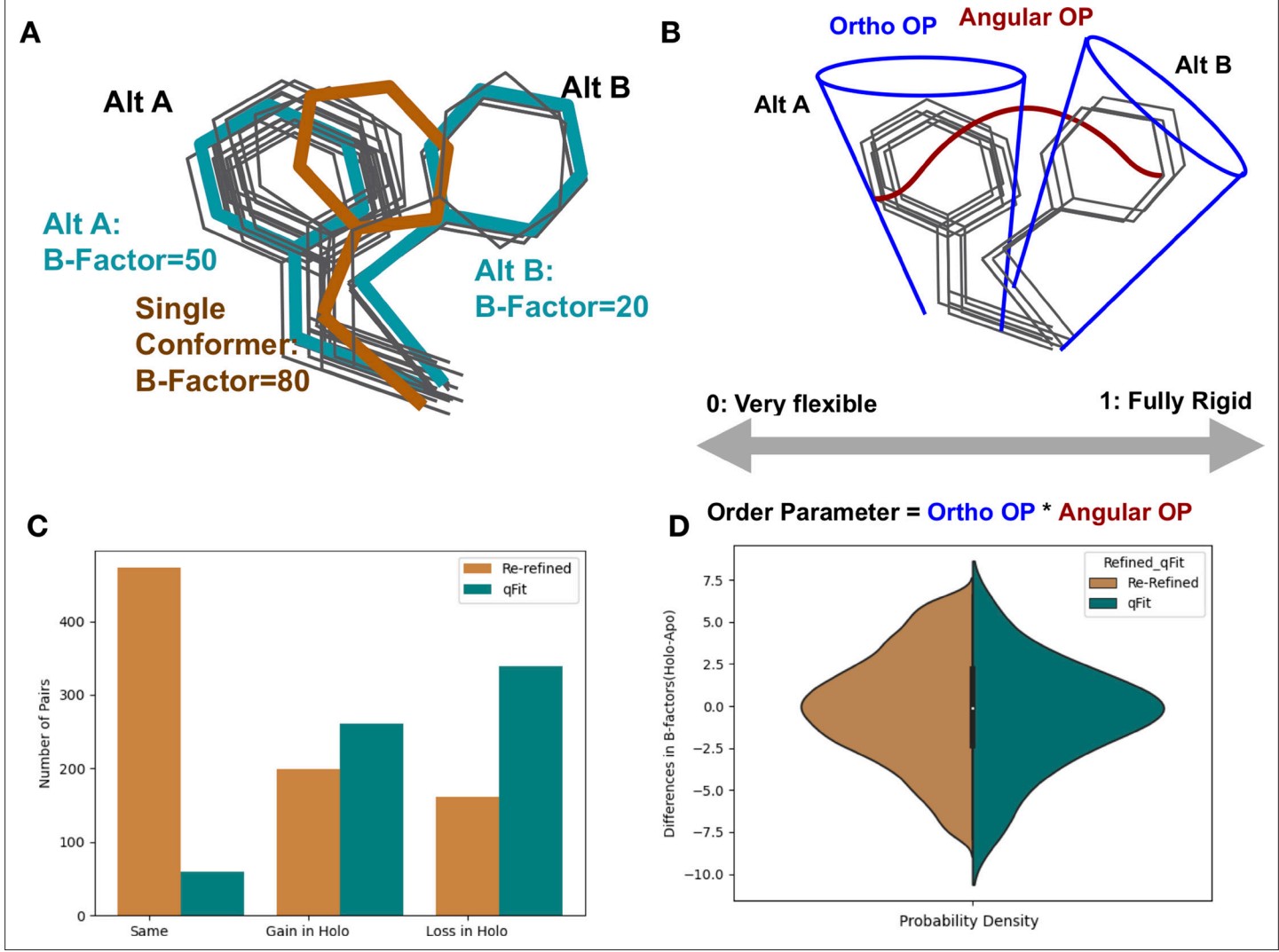

**Figure 1.** Representing structural data as multiconformer models. (**A**) The gray outlines represent snapshots of the true underlying ensemble of the phenylalanine residue. The orange stick represents the residue modeled as a single conformer. The teal sticks represent the residue modeled as alternative conformers. The single conformer accounts for all heterogeneity in the B-factor, increasing the B-factor and reducing our ability to determine harmonic versus anharmonic motion. When a residue is modeled using alternative conformers, this heterogeneity is divided between harmonic heterogeneity, captured by the B-factors of each alternative conformation and the anharmonic heterogeneity, captured by spread in coordinates between the alternative conformations. (**B**) To quantify the conformational heterogeneity of each residue, we used multi-conformer order parameters (*Fenwick et al., 2014*), which are the products of the ortho order parameter that captures the harmonic or B-factor portion of each conformation and the angular order parameter that captures the anharmonic portion or the displacement between alternative conformers. These are multiplied to produce the final order parameter (Materials and methods). (**C**) The change in the number of alternative conformers (holo-apo) in binding site residues. In the re-refined dataset (orange), the majority of structures have the same number of alternative conformers in the binding site, with the second most popular category gaining alternative conformers in the holo structure. In the qFit dataset (teal), the majority of structures lose an alternative conformer in the holo structure, with the second most common category being gaining an alternative conformer. (**D**) The differences in B-factors (holo-apo) in the re-refined (orange) and qFit (teal) datasets. Overall, there was no significant difference in B-factors between holo and apo structures in both the re-refined and qFit datasets.

The online version of this article includes the following figure supplement(s) for figure 1:

**Figure supplement 1.** Pipeline to create apo/holo pairs.

**Figure supplement 2.** Quality control of apo/holo pairs.

**Figure supplement 3.** Resolution distributions in apo/holo pairs.

**Figure supplement 4.** Ligand and protein type distributions in apo/holo pairs.

**Figure supplement 5.** Changes in the number of alternative conformations and B-factors in apo/holo pairs.

**Figure supplement 6.** Differences in B-factors between apo/holo pairs.

from NMR and improve our understanding of the energetics of binding (*van den Bedem and Fraser, 2015*).

Here, we examine how protein side-chain conformational heterogeneity changes upon ligand binding by assembling a large, high-quality dataset of matched holo and apo X-ray crystallography structures. To integrate both harmonic and anharmonic disorders, we use a consistent multiconformer modeling procedure, qFit (*Riley et al., 2021a*), and crystallographic order parameters (*Fenwick et al., 2014*). We test the hypothesis that ligand binding narrows the conformational ensemble, resulting in a decrease in heterogeneity of side chains in the holo structure compared with the apo structure. Our analysis reveals complex patterns of conformational heterogeneity that vary between and within proteins upon ligand binding. Specifically, in proteins where binding site residues become more rigid upon ligand binding, distant residues tend to become less rigid. This observation suggests that both natural and artificial ligands can modulate the natural composition of the protein conformational heterogeneity across the entire receptor to modulate the free energy of binding.

## Results
### Assembling the dataset
To assess the differences in conformational heterogeneity upon ligand binding, we identified high-quality, high-resolution (2 Å resolution or better) X-ray crystallography datasets from the PDB (*Berman et al., 2000*). We classified structures as holo if they had a ligand with 10 or more heavy atoms, excluding common crystallographic additives (*Supplementary file 1*, *Figure 1—figure supplement 1A*). Structures without ligands, excluding common crystallographic additives, were classified as apo (*Figure 1—figure supplement 1A*). We identified apo/holo matched pairs by requiring the same sequence and near-isomorphous crystallographic parameters. Furthermore, we required the resolution difference between holo and apo pairs to be 0.1 Å or less, selecting representative apo structures to minimize the difference in resolution (*Figure 1—figure supplement 1B*). This stringently matched ligand holo-apo dataset contained 1205 pairs (*Supplementary file 2*). We also used identical selection criteria to create a control dataset of 293 apo-apo pairs, taken from the set of apo/holo pairs (*Supplementary file 3*).

### Re-refining and qFit modeling of apo/holo pairs
To minimize biases resulting from different model refinement protocols, we re-refined all structures using the deposited structure factors and *phenix.refine* (*Liebschner et al., 2019*). The majority of structures in our re-refined dataset had less than 2% of residues modeled with alternative conformations, likely reflecting undermodeling of conformational heterogeneity represented in the PDB, based on prior literature (*Lang et al., 2010*). To more consistently assess conformational heterogeneity, we rebuilt all structures using qFit, an automated multiconformer modeling algorithm (*Keedy et al., 2015*; *Riley et al., 2021a*) with subsequent refinement using *phenix.refine* (*Liebschner et al., 2019*). While qFit has biases, running all models through a consistent protocol will avoid manual biases that could creep into the holo or apo structures specifically. Additionally, by re-building each model as a multiconformer model, we were able to better distinguish the contributions of harmonic and anharmonic conformational heterogeneity across the structure (*Figure 1A and B*). All models went through additional quality control, removing structures that resulted in large increases in R-free at each refinement step, high clashscores, or large root mean squared difference (RMSD) between the pairs (Materials and methods, *Figure 1—figure supplement 2*). This procedure resulted in 743 pairs. Due to apo datasets serving as the reference state for multiple ligand-bound structures, our dataset consists of 743 unique holo structures and 432 unique apo structures.

### Properties of the apo/holo pairs
The median resolution across our dataset was 1.6 Å with a small trend towards improved (higher) resolution in the apo structure (0.01 Å, median improvement [holo-apo]; p=3.8 × $10^{-20}$, Wilcoxon signed-rank test; *Figure 1—figure supplement 3A and B*). Across the dataset, 546 unique ligands were present in the structures, with 134 of these (e.g., NAG, AMP) appearing in multiple structures (*Figure 1—figure supplement 4A*). The median number of ligand heavy atoms was 19, with only 10 very large ligands (>50 heavy atoms, e.g., atazanavir; *Figure 1—figure supplement 4B*). The proteins

in the dataset represent 315 unique UniProt IDs, with a bias towards enzymes that have been used for model systems for structural biology, including endothiopepsin (n = 73 pairs), lysozyme (n = 62 pairs), trypsin (n = 48 pairs), and carbonic anhydrase 2 (n = 46 pairs; *Figure 1—figure supplement 4C and D*).

## Conformational heterogeneity across the re-refined and qFit dataset

To determine the differences in conformational heterogeneity upon ligand binding in both the re-refined and qFit models, we assessed four commonly used metrics: the number of alternative conformers, B-factors (atomic displacement parameter), root mean square fluctuations (RMSF), and rotamer changes.

### Number of alternative conformations

Alternative conformations were modeled at low frequency in the re-refined dataset compared to the qFit modeled structures (1.7% vs. 47.8% of residues). In the re-refined dataset, there is a bias to increased modeling of alternative conformations in the holo dataset (50.5% gain vs. 29.8% loss), whereas more even representation was observed in the qFit dataset (44.3% gain vs. 54.8% loss; *Figure 1—figure supplement 5A*). These results suggest that the trend of increased side-chain conformational heterogeneity in PDB deposited structures may have its origin in human bias with more careful human attention to careful model building of binding site residues in holo structures.

We next focused our analysis on binding site residues, defined as any residue with a heavy atom within 5 Å of any ligand heavy atom. In the re-refined dataset, 23.9% of the matched pairs had a gain in alternative conformations in the holo model compared to only 19.3% losing an alternative conformer in the holo model, suggesting, counterintuitively, that ligand binding increases local side-chain mobility (*Figure 1C*). However, in the qFit dataset, holo models tend to lose alternative conformations in the binding site residues (39.7% gain vs. 51.5% loss; *Figure 1C*).

### B-factors

Next, we explored the harmonic contribution to conformational heterogeneity as modeled by B-factors on a pairwise, residue-by-residue basis. Across all residues in the re-refined dataset, B-factors were slightly higher in holo models (0.31 Å$^2$, median difference [holo-apo]; p=4.4 × 10$^{-208}$, Wilcoxon signed-rank test; *Figure 1—figure supplement 5B*). In the qFit dataset, similar to the re-refined structures, holo residues had slightly higher B-factors (0.34 Å$^2$, median difference [holo-apo]; p=5.6 × 10$^{-264}$, Wilcoxon signed-rank test; *Figure 1—figure supplement 6A*). Of note, the B-factors in the qFit dataset are slightly smaller than the re-refined dataset (13.41 Å$^2$ vs. 13.94 Å$^2$, average B-factors), reflecting the tendency for alternative conformation effects to be modeled as increased B-factors. When examining the binding site residues, there was no significant difference in B-factors between the holo and apo models in both the re-refined (0.01 Å$^2$, median difference in B-factors; p=0.34, Wilcoxon signed-rank test; *Figure 1D*) and qFit datasets (0.06 Å$^2$, median difference in B-factors; p=0.7, Wilcoxon signed-rank test; *Figure 1D*, *Figure 1—figure supplement 6B*). The lack of change in B-factors close to ligands between the holo and apo models indicates that changes between the holo and apo B-factors are driven by signals distant from the binding site.

## Conformational differences incorporating alternative conformations

Because of the low number of alternative conformers in the re-refined dataset, we only explored the anharmonic differences for side chains between the holo and apo models in the qFit dataset. First, to determine the extent of conformational change of alternative conformations, we compared the rotameric distribution of side chains. Side-chain rotamer changes between apo and holo structures have been reported to be very prevalent in single-conformer models, with 90% of binding sites having at least one residue changing rotamers upon ligand binding (*Gaudreault et al., 2012*). To accommodate multiconformer models, we assigned all conformations to distinct rotamers using *phenix.rotalyze*. We classified each residue as having 'no change' in rotamers if the set of rotamer assignments matched the holo and apo residue. In binding sites, we also observed that 'no change' was the most common outcome for residues (78.6%; *Figure 2A*). In the second largest category, 'distinct,' the holo and apo residues shared no rotamer assignments (15.5% of residues; *Figure 2B*).

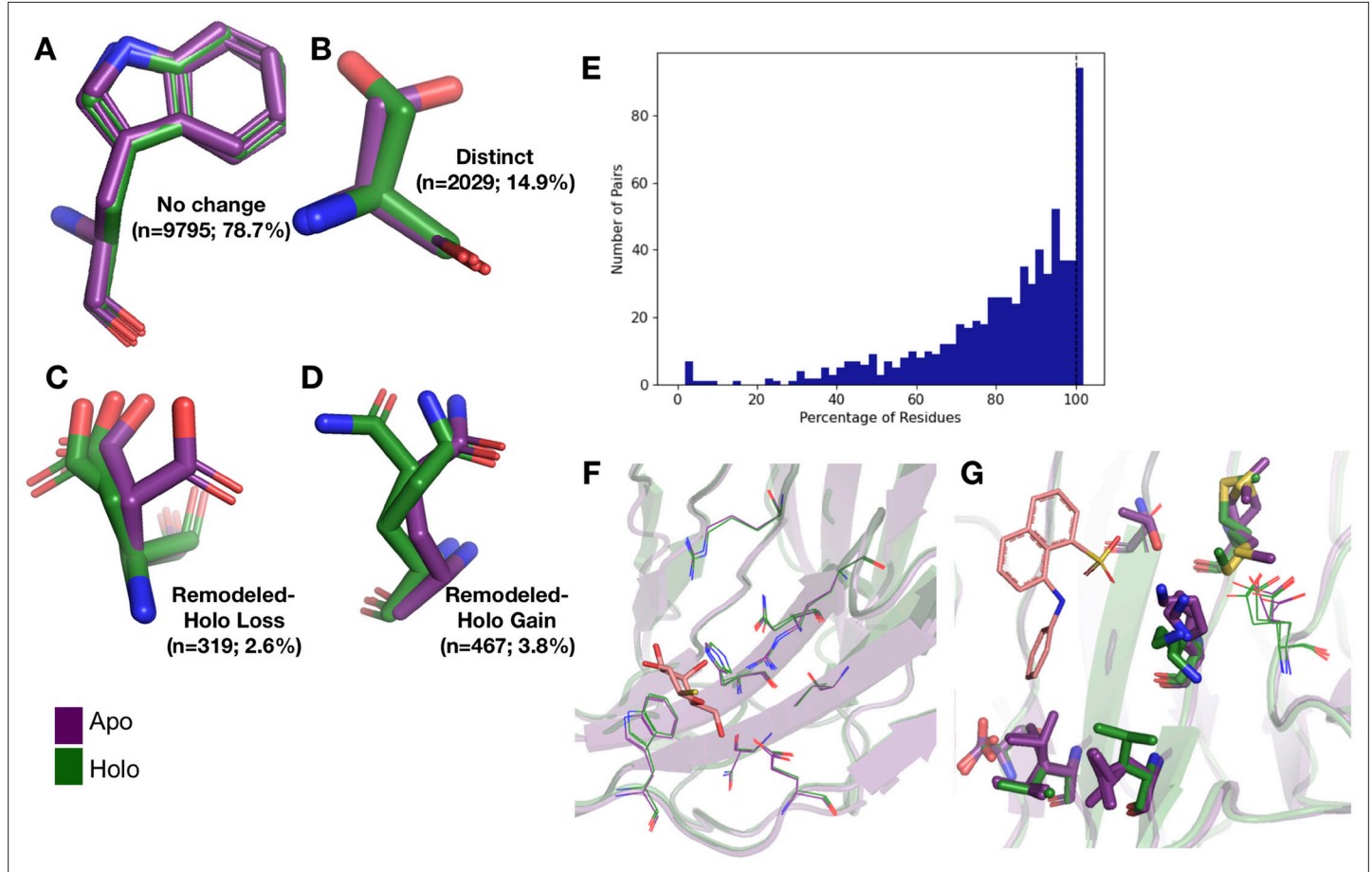

**Figure 2.** Examples of rotamer changes between apo (purple) and holo (green) binding site residues. (**A**) Example residues for: 'no change' in rotamer status, accounting for 78.7% of binding site residues; (**B**) 'distinct' rotamers, accounting for 14.9% of binding site residues; (**C**) 'remodeled-holo loss,' accounting for 2.6% of binding site residues; and (**D**) 'remodeled-holo gain,' accounting for 3.8% of binding site residues. (**E**) The percentage of residues in the binding site that have the same rotamer status in the holo and apo structures. The black line highlights the 11% of pairs that had the same rotamer status for all binding site residues. (**F**) Paired galectin-3 apo (purple; PDB: 5NFC) and holo (green; PDB: 4JC1, ligand: thiodigalactoside) multiconformer models with no changes in rotamer status in any binding site residues. (**G**) Paired transthyretin apo (purple; PDB: 1ZCR) and holo (green; PDB: 3CFN, ligand: 1-anilino-8-naphthalene) multiconformer models with six out of nine residues with remodeled or different rotamer status in the binding site residues. Residues with rotamer changes are shown as sticks. Residues with no change in rotamer status are shown as lines.

The online version of this article includes the following figure supplement(s) for figure 2:

**Figure supplement 1.** Differences in RMSF between apo/holo pairs.

A more complicated situation occurs when some, but not all, of the rotamer assignments are shared across apo and holo residue. We classified 2.6% of residues as 'remodeled-holo loss' (***Figure 2C***) if distinct, additional rotameric conformations were populated in the apo residue only and 3.8% of residues as 'remodeled-holo gain' (***Figure 2D***) if distinct, additional rotameric conformations were populated in the holo residue only. These results suggest a counterintuitive interpretation of binding site residues increasing their conformational heterogeneity upon ligand binding. However, a major potential confounder is that holo structures reflect an ensemble average of two compositional states (apo and holo) with alternative conformations representing the apo state at reduced occupancy, which we examined by subsetting the ligands based on relative B-factors (see below). A potential for a third category of remodeling, where both apo and holo residues share at least one conformation and each have at least one additional conformation, did not occur in our dataset.

Across apo-holo pairs, there was a large range of the percentages of binding site residues with the same rotamer classification in the pairs (23.2–100.0%), indicating that side-chain remodeling can be quite variable (***Figure 2E***). We found that 11% of binding sites had all residues classified as 'same' between pairs, consistent with a previous study that used single-conformer models (***Gaudreault***

*et al., 2012*). As an example of such a 'pre-organized' binding site is galectin-3 bound to thiodigalac-toside (PDB: 5NFC, 4JC1; *Figure 2F*). In contrast, 67% of binding site residues have a rotamer status difference in transthyretin (PDB: 1CZR, 3CFN; *Figure 2G*), including a rotamer change in Leu101 to avoid a clash with the ligand.

To compare the magnitude of fluctuations between alternative conformations, we calculated RMSF for all residues. This analysis suggested that, on average, apo residues have slightly greater conforma-tional heterogeneity than holo residues (–0.006 Å, mean difference of RMSF(holo-apo); p=3.7 × 10⁻⁸, Wilcoxon signed-rank test; *Figure 2—figure supplement 1A*). This trend was somewhat stronger in binding site residues (–0.02 Å, mean difference of RMSF(holo-apo); p=4.5 × 10⁻²⁹, Wilcoxon signed-rank test; *Figure 2—figure supplement 1B*). Our RMSF results suggest that, on average, there is a slight decrease in heterogeneity upon ligand binding and that this reduction is most prevalent at residues distant from the binding site.

Collectively, these results do not conform to a simple model. There is a large amount of variability in the response across datasets and the median responses reveal only small biases. Nonetheless, considering those average responses, upon binding a ligand, the RMSF analysis suggests decreases in heterogeneity at the binding site, whereas the rotamer comparison has a slight bias to increased heterogeneity at the binding site, and B-factors only change at distant sites. One interpretation is that heterogeneity is reduced in binding site residues by a small number of anharmonic conformational changes, as observed by the RMSF reduction, paired with an increase in harmonic fluctuations far away, as observed by an increase in the B-factors. However, it is difficult to interpret these changes separately as conformational heterogeneity is a combination of both harmonic and anharmonic motion and there is potential degeneracy in modeling alternative conformations, even with qFit (*van den Bedem et al., 2009*). Therefore, we moved to using an integrated measurement of order param-eters that can account for these complications (*Fenwick et al., 2014*).

## Order parameters integrate both harmonic and anharmonic conformational heterogeneity

To integrate the anharmonic fluctuations between alternative conformers with the harmonic fluctu-ations modeled by B-factors (*Kuzmanic et al., 2014*), we used a crystallographic order parameter (*Figure 1A*; *Fenwick et al., 2014*). Order parameters allow us to capture the conformational entropy both within and between side-chain rotamer wells. While order parameters are traditionally used in NMR or molecular dynamic simulations, they can be calculated for multiconformer X-ray models and, in some cases, show reasonable agreement with solution measures (*Fenwick et al., 2014*). We focused on the order parameters of the first torsion angle ($\chi 1$) of every side chain for all residues except for glycine and proline. Order parameters are measured on a scale of 0–1, with 1 representing a fully rigid residue and 0 representing a fully flexible residue. Below, we analyze the differences in normalized order parameters between paired residues (Materials and methods, *Figure 3—figure supplement 1*).

As an additional control, we compared our apo/holo dataset to a dataset of apo/apo pairs. In examining the differences in order parameters, both in the apo/holo pairs and the apo/apo pairs, there are no large differences in conformational heterogeneity, as indicated by a median difference in order parameters of approximately 0. However, in the apo/holo pairs there is a much wider range of order parameter differences, indicating that ligand binding impacts conformational heterogeneity beyond experimental variability (p=3.4 × 10⁻¹⁷, individual Mann–Whitney *U* test; *Figure 3A*). We also explored the different binding site cutoff values, ranging from 2 to 10 Å, observing that the tighter the binding site definition, the more drastic the difference in order parameters between holo and apo pairs (*Figure 3—figure supplement 2*).

Next, to examine whether different regions of the protein were driving this higher variability, we compared the differences in order parameters among binding site residues, within 5 Å of any ligand heavy atom, compared to a control dataset that matched the number of, type and solvent exposure within the protein for each binding site residue. In binding site residues, the holo structures had a slightly, but significantly, increased order parameters, suggesting reduced conformational heteroge-neity compared to the control dataset (0.034 vs. 0, median difference [holo-apo] order parameter; p=3.4 × 10⁻⁷, individual Mann–Whitney *U* test; *Figure 3B*). While there is a larger range of responses, this indicates that, in general, binding site residues become more rigid upon ligand binding.

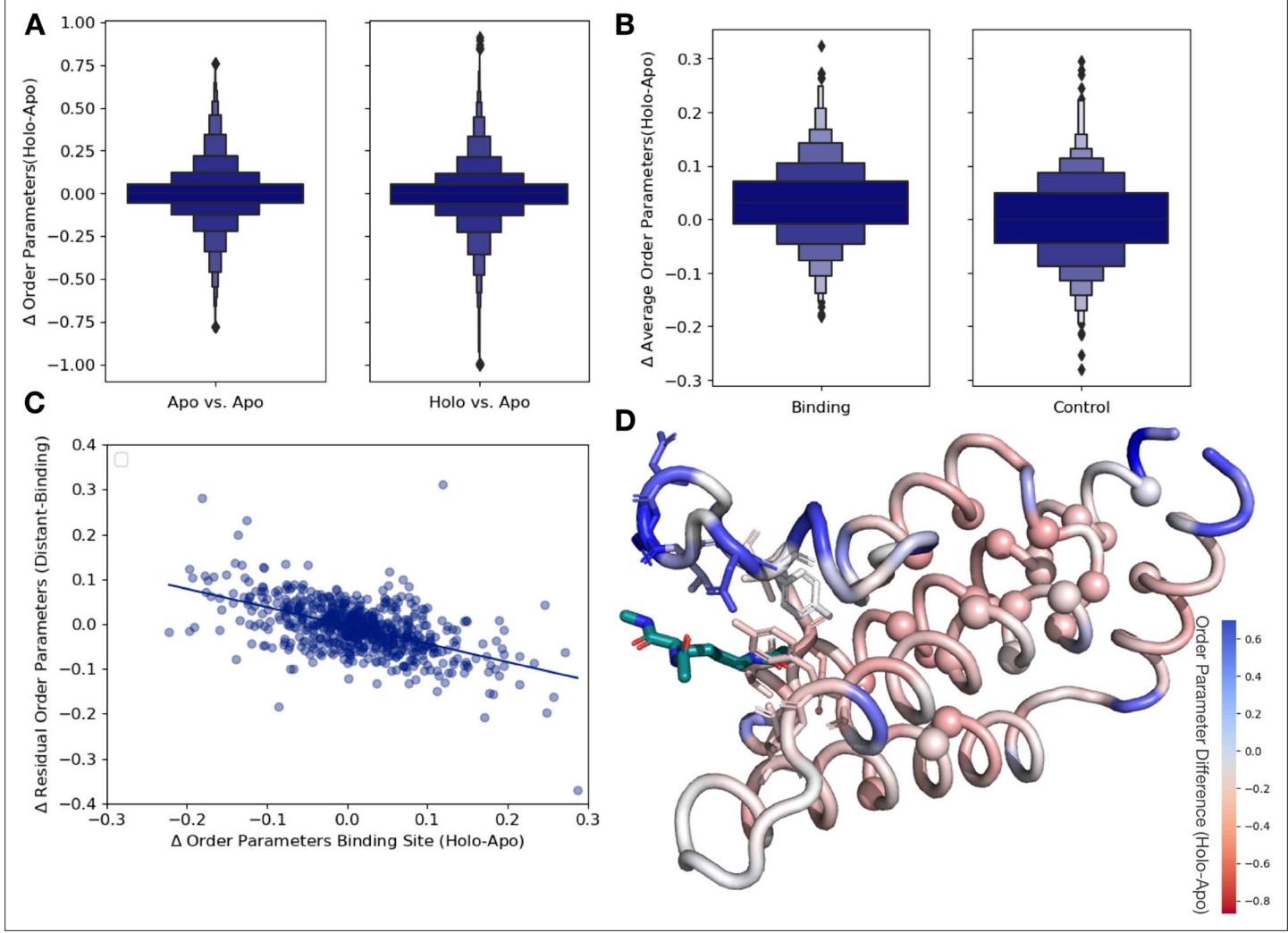

**Figure 3.** Ligand binding alters conformational heterogeneity patterns. (**A**) Across all residues, the distribution of order parameter changes is much wider in holo-apo pairs compared to apo-apo pairs (p=3.4 × 10⁻¹⁷, individual Mann–Whitney *U* test); however, there is no median difference in order parameters upon ligand binding (median difference: 0 for both), indicating that ligands have varying impacts across different proteins. (**B**) The distribution of the average differences of order parameters in binding site residues compared to the average differences in a control dataset made up of the same number, type, and solvent exposure of amino acids. Comparing the apo/holo structures, on average binding site residues got more rigid upon binding. The median difference in order parameters was 0.03 for the binding site residues compared to 0 for the control dataset (p=3.4 × 10⁻⁷, individual Mann–Whitney *U* test). (**C**) The relationship of the difference in order parameters between the holo and apo residues in binding site residues versus the residual order parameter in distant, non-solvent-exposed residues. We observed a negative trend (slope = −0.44), indicating that structures that had a loss of heterogeneity in the binding site (right on the x-axis) had a relative gain in heterogeneity in residues distant from the binding site that were not solvent exposed (top on the y-axis). (**D**) We explore this trend in a structure of human ATAD2 bromodomain (PDB: 5A5N). Residues are colored by the differences between the average binding site order parameter minus the order parameter for each residue. Blue residues are less dynamic than the average binding site residue, and red residues are more dynamic than the average binding site residue. Binding site residues are represented by sticks, and distant, non-solvent-exposed alpha carbons are represented by spheres. The ligand ((2S)-2,6-diacetamido-*N*-methylhexanamide) is colored in teal.

The online version of this article includes the following figure supplement(s) for figure 3:

**Figure supplement 1.** To normalize the order parameters across all structures, we looked at 31 lysozyme structures and compared their order parameters.

**Figure supplement 2.** Distribution of order parameters differences at different binding site cut-offs.

**Figure supplement 3.** Order parameter relationships.

**Figure supplement 4.** X-ray versus NMR order parameters in HEWL.

**Figure supplement 5.** The differences in hydrogen bonds across all binding site residues.

## Spatial distribution of conformational heterogeneity changes

Based on the large range of order parameter differences we observed across the protein, along with the decrease in heterogeneity localized to binding site residues, we next explored the relationship between changes in heterogeneity in binding site residues and the rest of the protein. The difference in order parameters between the holo and apo models was correlated in both the binding site and distant residues (*Figure 3—figure supplement 3A*), indicating that ligand binding generally caused global changes to flexibility. Given the average rigidification of the binding site residues (*Figure 3B*, *Figure 3—figure supplement 3B*), these results predict a general trend of decreased conformational heterogeneity in the ligand binding site would be associated with a relative increase in conformational heterogeneity at distant sites in the protein. This pattern suggests that the residual change in heterogeneity (the difference between the average order parameter of the distant residues and the average order parameter of the binding site residues) should be inversely related to the change in the binding site residues: more rigidified binding sites will have more flexible than expected distant sites, and vice versa. Therefore, we explored the relationship between binding site residues and distant residues, defined as those more than 10 Å away from any heavy atom in the ligand. Indeed, on a protein-by-protein basis, the relationship between binding site residues and residual changes at distant sites follows this trend (*Figure 3—figure supplement 3C*). We also explored if protein size impacts our results, but did not observe any trend between protein size and order parameter correlation (*Figure 3—figure supplement 3E*). Consistent with studies suggesting significant residual conformational heterogeneity in folded buried residues (*Wong and Daggett, 1998*) and the potential for those buried residues to change heterogeneity upon ligand binding (*Moorman et al., 2012*), this trend is even stronger in residues that were more than 10 Å away from any heavy atom in the ligand and less than 20% solvent exposed (slope = −0.44, $r^2$ = 0.46; p=5.1 × 10⁻⁵⁰, two-sided *t*-test; *Figure 3C*). This indicates that proteins that lose conformational heterogeneity in the binding site are associated with a relative increase in conformational heterogeneity in distant, non-solvent-exposed residues.

There are three likely origins of this effect. First, this may reflect a feature of the distribution of order parameters around the mean value within each protein. Second, this may reflect a topological feature of protein packing, whereby packing optimization of certain areas of a protein decreases the optimization of other parts of the protein (*Bromberg and Dill, 1994*). Third, this may reflect the stabilization of certain conformations in a ligand-bound protein. As a control for these effects, we compared the residual order parameter differences between the buried, non-solvent-exposed residues and the binding site residues in apo-apo pairs. Globally the trends were similar, but weaker in both correlation and magnitude (slope = −0.28, $r^2$ = 0.20; p=1.8 × 10⁻³⁴, two-sided *t*-test; *Figure 3—figure supplement 3D*). The difference in the slope between the holo-apo and apo-apo dataset was further compared using a bootstrap analysis, demonstrating that the mean slope of the holo-apo is more than 2 standard deviations away from the apo-apo slope, representing the robustness in differences between the two slopes (p=0.0, z-test; *Figure 3—figure supplement 3F*). Therefore, we interpret the trend we observe as mainly based on protein topology, specifically that proteins have areas where there are less efficiently packed alternative conformers, likely to enable entropic compensation across the protein during various functions, including ligand binding. We interpret that the stronger signal we observed in the holo-apo dataset is due to the ligand perturbation, which is also reflected in the median rigidification of binding site residues (*Figure 3B*). We hypothesize that we are observing this innate protein property being used, specifically optimizing the binding site residues to bind a ligand, while decreasing the optimization elsewhere in the protein.

As an example to visualize this trend, we mapped the change in order parameters onto the structure of the human ATAD2 bromodomain (PDB ID: 5A5N). In ATAD2, the binding site residues rigidify upon ligand binding, whereas the majority of distant residues are more heterogeneous compared to the binding site residues (*Figure 3D*). Specifically, this difference is greatest between binding residues and non-solvent-exposed residues, as previously observed in lysozyme; however, there was only weak residue to residue correlation (*Figure 3—figure supplement 4A and B*; *Moorman et al., 2012*). However, as in the global analysis, the ATAD2 example demonstrates there is a large range of changes in binding site order parameters, consistent with NMR examples that show a heterogeneous response both close to and distant from ligands (*Caro et al., 2021*).

## Hydrogen bond patterns change upon ligand binding

We next investigated changes in protein side-chain hydrogen bonds upon ligand binding. Here, we applied HBplus (*McDonald and Thornton, 1994*) to identify hydrogen bonds for each side-chain alternative conformation (Materials and methods). We examined the occupancy-weighted hydrogen bonds in binding site residues using a hydrogen bond cutoff of 3.2 Å. Overall, we observed the creation of 0.06 hydrogen bonds per residue in holo binding sites (*Figure 3—figure supplement 3*), which translates to 10% of structures gaining one full hydrogen bond in the holo structure. This is likely indicative of stable binding sites in holo structures. This follows a trend observed previously where upon ligand binding, hydrogen bonds to water molecules decrease, but hydrogen bonds to other protein atoms increase (*Gaudreault et al., 2012*).

## Ligand properties influence conformational heterogeneity

Next, we investigated how ligand properties impact the conformational heterogeneity of binding site residues. For ligand properties dictated by the size of the ligand (number of rotatable bonds and number of hydrogen bonds), we normalized these metrics by the molecular weight of the ligand. For each property, we compared the highest and lowest quartiles by both the absolute order parameters of the holo structure and the order parameter differences between holo and apo pairs. No significant associations existed when comparing the differences between holo and apo order parameters, but the characteristics of the holo binding site and the rotamer changes were correlated with ligand properties in several cases.

We hypothesized that ligand properties associated with increased ligand dynamics, including more rotatable bonds, higher lipophilicity (logP), fewer hydrogen bonds, and more heavy atoms would be associated with increased conformational heterogeneity (an increase in absolute order parameters or a smaller difference between the apo and holo order parameters; *Wicker and Cooper, 2015*). While molecules with fewer rotatable bonds (lower quartile: <2 [n = 134] vs. upper quartile: >6 [n = 134]) were indeed associated with more rigid binding sites (lower quartile: 0.83 vs. upper quartile: 0.81, individual Mann–Whitney $U$ test), this was not significant. However, higher numbers of rotatable bonds were associated with a lower number of same rotamers between the apo and holo binding site residues (88% vs. 80%, percentage same rotamer; p=6.0 × 10⁻⁶, individual Mann–Whitney $U$ test; *Figure 4—figure supplement 1*). Increased lipophilicity (logP, upper quartile: <0.04 [n = 134] vs. lower quartile: >2.69 [n = 134]) was significantly associated with a more flexible binding site (0.79 vs. 0.84, median order parameters; p=7.5 × 10⁻⁶, individual Mann–Whitney $U$ test; *Figure 4A*). Previous studies have indicated that increased lipophilicity generates more nonspecific binding interactions (*Olsson et al., 2008*). Larger compounds (upper quartile: >26 heavy atoms [n = 134] vs. lower quartile: <13 heavy atoms [n = 134]) are also associated with more flexible binding sites (0.83 vs. 0.79, median order parameter; p=0.0001, individual Mann–Whitney $U$ test; *Figure 4B*). Large compounds, thus a larger ligand surface area, are associated with more nonspecific binding interactions, which is compatible with increased protein conformational heterogeneity. Finally, more total hydrogen bonds per heavy atom (upper quartile: >0.47 [n = 134] vs. lower quartile: <0.25 [n = 134]) are associated with more rigid binding sites (0.84 vs. 0.79, median order parameter; p=5.9 × 10⁻⁵, individual Mann–Whitney $U$ test; *Figure 4C*). This trend holds even when examining hydrogen bond donors or acceptors separately.

From these results, an intuitive general picture emerges where more specific, directional interactions, such as hydrogen bonds (*Bissantz et al., 2010*), are more likely to lock the corresponding protein residue in place, thus creating more rigid binding site residues (*Majewski et al., 2019*). Whereas the more nonspecific interactions are correlated with more flexible binding site residues. There is also a wide range of deviation from this general picture, likely reflecting that natural and artificial optimization of ligands is based on free energy, not any specific thermodynamic component or interaction type. These trends emphasize the need to monitor both the impacts of ligands on specific interactions with the protein along with conformational heterogeneity of the protein. Additionally, these results suggest that specific interactions can be tuned to rigidify a binding site. Paired with our findings of the relationship between order parameters in binding site and distant residues, ligand impacts are likely propagated throughout the protein. Ligands with more specific interactions, thus a less flexible binding site, will likely have a corresponding increase in conformational heterogeneity distant from the binding site.

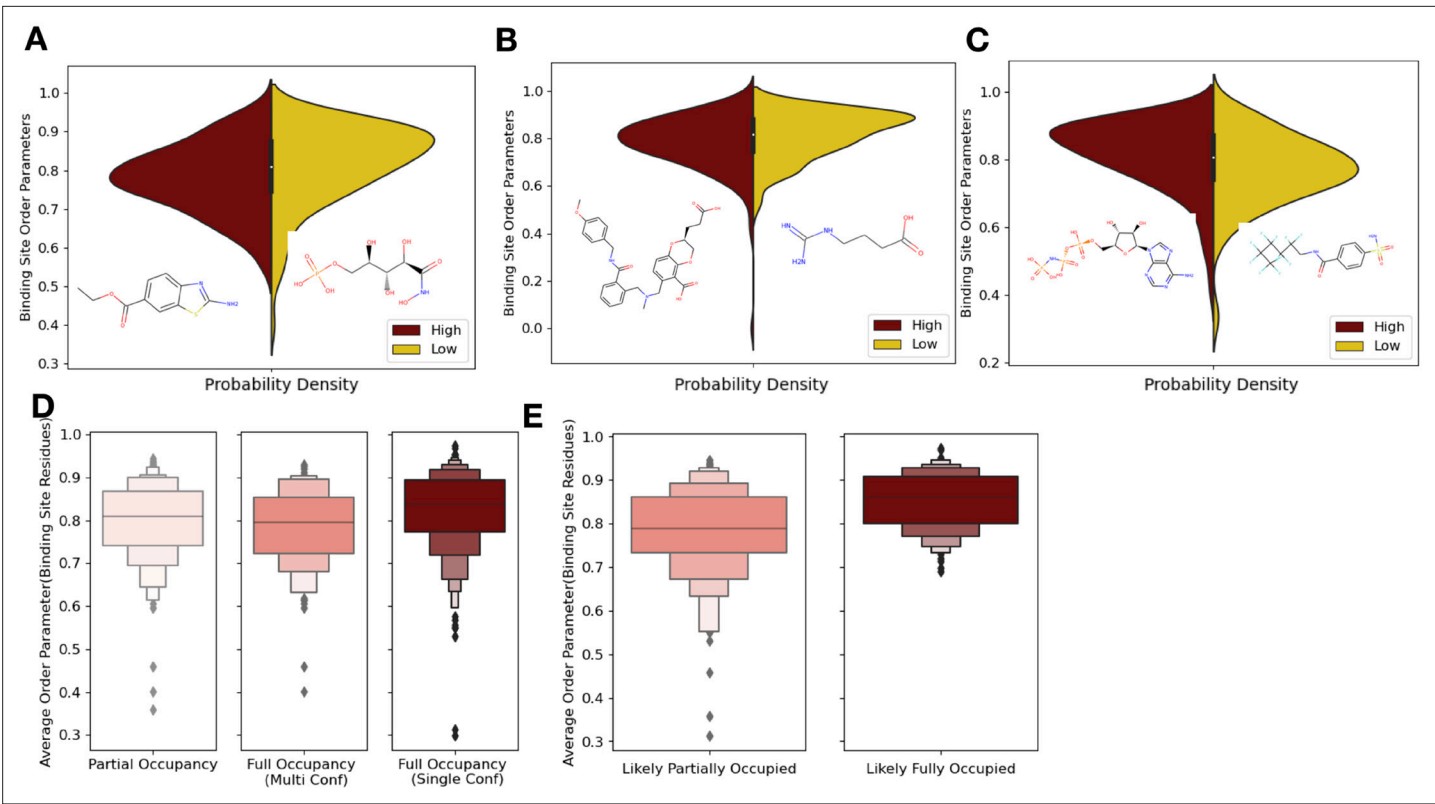

**Figure 4.** Ligand properties impact binding site order parameters. (**A**) Ligands with higher logP value (maroon), indicative of more greasy or hydrophobic ligands, versus ligands with a lower logP value (gold), had lower in order parameters in the binding site residues (0.78 vs. 0.84, median order parameter; p=7.5 × 10⁻⁶, independent Mann–Whitney *U* test) (example ligands: low logP: 5-phospho-D-arabinohyroamic acid; high logP: ethyl 2-amino-1,3-benzothiazole-6-carboxylate). (**B**) Ligands with relatively higher molecular weight (maroon) had higher-order parameters compared to those with lower molecular weight (gold; 0.79 vs. 0.83, median order parameter; p=0.0001, independent Mann–Whitney *U* test) (example ligands: high number of heavy atoms: (2S)-2-(3-hydroxy-3-oxopropyl)–6-[[[2-[(4-methoxyphenyl)methylcarbamoyl]phenyl]methyl-methyl-amino]methyl]-2,3-dihydro-1,4-benzodioxine-5-carboxylic acid; low number of heavy atoms: 4-carbamimidamidobutanoic acid). (**C**) Ligands with relatively higher hydrogen bonds per heavy atom (maroon) had higher-order parameters compared to those with lower molecular weight (gold; 0.84 vs. 0.79, median order parameter; p=5.9 × 10⁻⁵, independent Mann–Whitney *U* test) (example ligands: low hydrogen bond: 4-sulfamoyl-*N*-(2,2,3,3,4,4,5,5,6,6,6-undecafluorohexyl) benzamide; high hydrogen bond: phosphoaminophosphonic acid-adenylate ester). (**D**) Binding site order parameters were lower in ligands with partial occupancy (light pink; 0.79, median order parameter) and multiconformer ligands adding to full occupancy (salmon; 0.80, median order parameter) compared to single-conformer ligands with full occupancy (dark red; 0.83, median order parameter; p=4.9 × 10⁻⁸, independent Mann–Whitney *U* test). (**E**) In fully occupied ligands, ligands in the top quartile of ligand B-factors, controlled for by the mean alpha carbon B-factor, had lower binding site order parameters (salmon; 0.79, median order parameter) compared to ligands in the bottom quartile (dark red; 0.85, median order parameter; p=1.6 × 10⁻¹¹, independent Mann–Whitney *U* test).

The online version of this article includes the following figure supplement(s) for figure 4:

**Figure supplement 1.** Ligand properties relationship with order parameters.

## Reduced ligand occupancy and conformational heterogeneity

One potential confounder for quantifying the change in conformational heterogeneity of binding site residues is that the ligands may not be fully occupied in the crystal. There were 193 structures with ligands with alternative conformations or partially occupied ligands in our datasets (*Figure 4D*). Of these 193, 125 ligands had less than full occupancy, whereas 68 had alternative conformations that amounted to full occupancy. The vast majority of ligands (n = 425) were modeled originally with full occupancy. Fully occupied ligands were associated with more rigid binding sites than partially occupied ligands or ligands with alternative conformers (0.84 vs. 0.79, mean order parameters of binding site residues; p=2.9 × 10⁻⁷, individual Mann–Whitney *U* test; *Figure 4D*). There was no difference observed between the partially occupied ligands and ligands with alternative conformers (p=0.15, individual Mann–Whitney *U* test). We also explored if partially

occupied ligands were associated with more rotamer changes between holo and apo pairs, but no significant difference existed (80% vs. 85%, median percentage of the same rotamer; *Figure 4—figure supplement 1B*).

While the scattering contributions of B-factor and occupancy changes are subtle (but distinct), most models likely include true occupancy changes as elevated B-factors. We observed a wide range of average ligand B-factors and, as expected, a lack of correlation between the ligand B-factors and ligand occupancy (*van Zundert et al., 2018*; *Bhat, 1989*; *Carugo, 1999*). As a proxy for likely partially occupied ligands, we normalized the ligand B-factor by the mean C-alpha B-factor to identify ligands with higher B-factors than expected (*Figure 4—figure supplement 1C*). We examined the outer two quartiles of the normalized ligand B-factors (>0.016 vs. <0.005, median normalized B-factor). In these 'likely partially occupied' ligands, we observed greater conformational heterogeneity (0.86 vs. 0.80, mean order parameter; p=$1.6 \times 10^{-11}$; individual Mann–Whitney $U$ test, *Figure 4E*). In structures with modeled partially occupied ligands and likely partially occupied ligands, we learned that binding site residues tend to have more apparent conformational heterogeneity, likely due to a combination of compositional and conformational heterogeneity.

## Conformational heterogeneity for multiple ligands to CDK2

To better understand our findings in the context of multiple, diverse ligands binding to a single receptor, we examined cyclin-dependent kinase 2 (CDK2), a cyclin kinase family that regulates the G1 to S transition in the cell cycle. Our dataset contains 13 protein inhibitor complexes, including both type I and type II inhibitors, all of which share the same apo model (PDB ID: 1PW2). We hierarchically clustered the residues and ligands by difference in order parameters between the holo and apo models, identifying three distinct clusters of residues. The first cluster (blue, *Figure 5A and B*), consisting of 13 residues, is rigidified upon ligand binding. This cluster included residues scattered throughout both the N- and C-lobes of CDK2 that rigidified upon ligand binding. Two notable residues in this cluster, Glu127 and Val18, contact the inhibitors. Upon ligand binding, Val18 transitions from multiple conformers to a single conformation. Glu127 has a similar conformation in the apo and type II structures of two distinct alternative side-chain rotamers, whereas in the type I inhibitor structure, the alternative conformers cluster around the same rotamer (*Figure 5—figure supplement 1A and B*).

The second cluster (salmon, *Figure 5A and B*) consists of 14 residues that increase flexibility upon ligand binding. The majority of these residues connect the P-loop and the activation loop (*Figure 5B*). The electron density is very weak for many of these residues in most of the holo structures, driving their modeling in multiple conformations and elevated B-factors (*Figure 5—figure supplement 1C*). We also observed that many of these residues had side chain to side chain hydrogen bonds that were lost upon ligand binding (*Figure 5—figure supplement 2A*). The third cluster (dark red, *Figure 5A and B*) comprises five residues that became more flexible in all, but two holo datasets, which are the only type II inhibitors in the dataset. These were all located on the activation loop of the kinase (*Figure 5C*). As type II inhibitors, the two molecules (PDB: 1PXI [ligand: CK1] and PDB: 3QQL [ligand: X03]) bind the DFG out of conformation present in the apo dataset (PDB: 1PW2) and do not have as drastic a rigidifying effect as the type I inhibitors. Notably, these two inhibitors were also smaller than the type I inhibitors and the reduced contacts may also drive some of this effect. We also observed that the hydrogen bonds gained in the holo structure are inhibitor specific (*Figure 5—figure supplement 2B and C*).

The multiconformer models also provide a structural rationale for these changes. The differences in DFG conformation change the contacts with the P-loop, which allow for greater side-chain flexibility in the 'up' form compatible with type I inhibitors. The interface between the P-loop and the activation loop is weaker and residues such as Tyr155 adopt multiple conformations. At the base of the activation loop, Thr161, a critical phosphorylation site, changes conformation, with a rigidifying effect common to both type I and II inhibitors (*Figure 5D*, *Figure 5—figure supplement 2D*). The conformation of Thr161 found in the type II inhibitors overlaps, with one of the conformations populated in the multiconformer apo model. In contrast, the type I inhibitors adopt a distinct conformation. This case study highlights how modeling information present in the density can reveal changes beyond those in single-conformer structures.

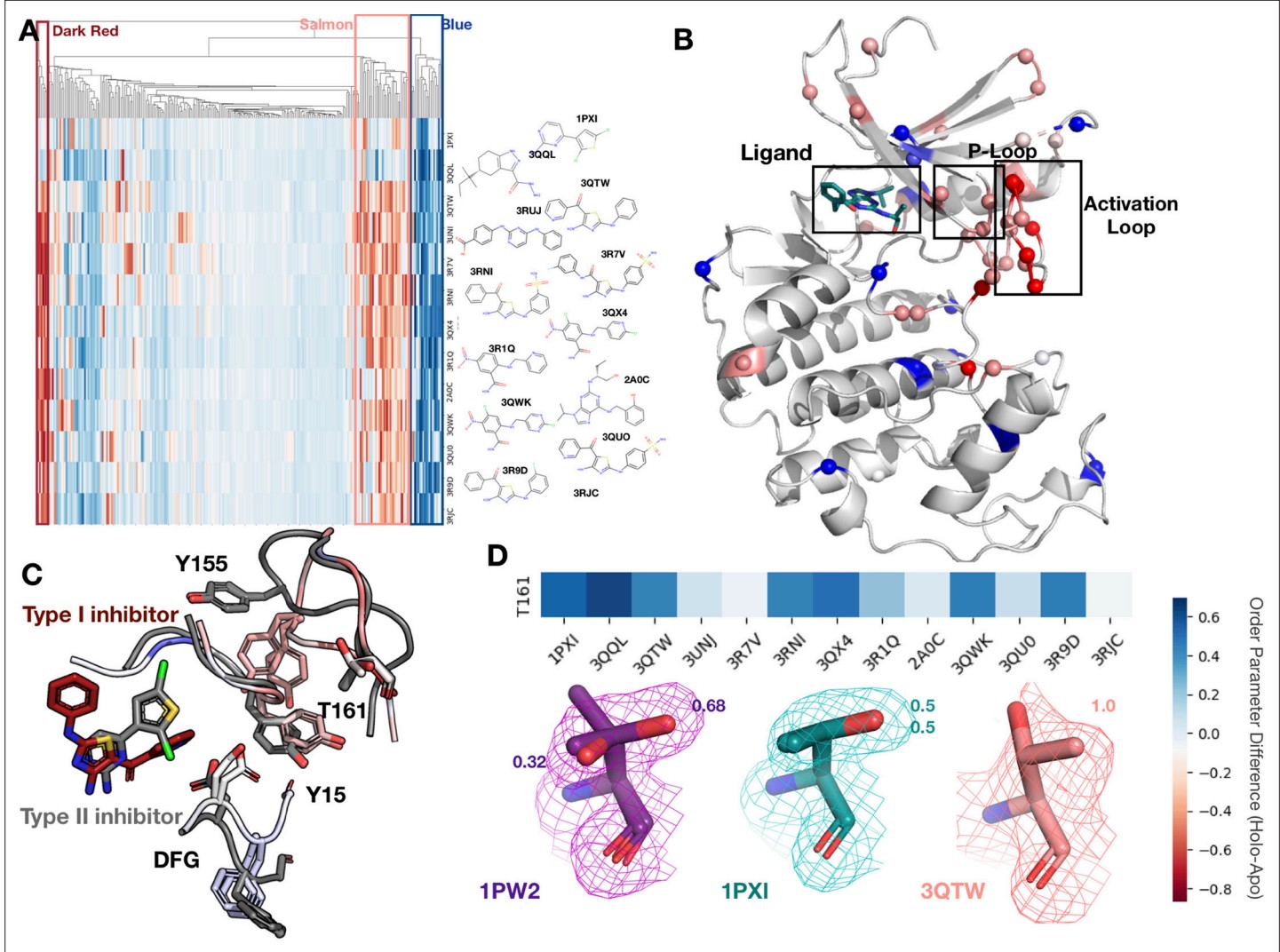

**Figure 5.** Conformational change and heterogeneity in CDK2. (**A**) The clustermap of all residues in the 13 CDK2 protein/ligand pairs. Red values indicate a negative difference (holo-apo) in order parameters, indicating that the holo structures have more heterogeneity compared to the apo. Blue values indicate positive differences, indicating that the apo structures have more heterogeneity compared to the holo. We highlighted three important clusters, the left red cluster, middle salmon cluster, and right blue cluster. (**B**) A representative structure (PDB: 3QTW) is shown with each residue colored by the difference in order parameter, corresponding to the same coloring scheme as the clustermap. The three distinct clusters (dark red, salmon, blue) are shown in spheres. (**C**) Many of the key differences between type I inhibitor (PDB: 3QTW) and type II inhibitor (PDB: 1PXI) are located in the DFG motif, P-loop, and activation loop. The type II inhibitor structure is colored in gray, and the type I inhibitor is colored as the difference in order parameters between the type I inhibitor and type II inhibitor structures. Red signifies a more dynamic region in the type I inhibitor structure, and blue signifies a less dynamic region in the type I inhibitor structure. Changes in the DFG motif, propagates changes, both structural and in dynamics, in the P-loop (highlighted by Tyr15), which propagates even larger changes in the activation loop between the two inhibitors, including changes in conformation of Thr161, the phosphorylation site of CDK2. (**D**) Threonine 161, the phosphorylation site for CDK2. We looked at the supporting density for specific residues between the apo (PDB: 1PW2, purple), type II (PDB: 1PXI, teal), and type I (PDB: 3QTW, salmon) inhibitors. 2Fo-Fc electron density is shown at 1 sigma. Occupancies of the alternative conformers are labeled with the corresponding color. The apo structure has multiple conformations, whereas the type I model only has one, and the type II model has two very similar conformations, but these are in different rotamer states compared to the apo.

The online version of this article includes the following figure supplement(s) for figure 5:

**Figure supplement 1.** We looked at the difference in order parameters (holo-apo) and the supporting density for specific residues between the apo (PDB: 1PW2, purple), type II (PDB: 1PXI, 3QQL, teal), and type I (PDB: 2A0C, 3QTW, 3R1Q, salmon) inhibitors.

**Figure supplement 2.** Differences in hydrogen bonds across CDK2 structures.

## Discussion

By creating a large dataset of stringent matched pairs of apo and holo multiconformer models, we identified a pattern of conformational heterogeneity consistent with smaller-scale studies of individual proteins (*Caro et al., 2021*). We observed that individual proteins greatly varied in amount and direction of change of conformational heterogeneity, as observed in previous studies (*Caro et al., 2017*). In general, we found that binding site residues tend to become more rigid upon ligand binding. But similar to the entire protein, there was a large range of effects, including many sites becoming more flexible when bound to a ligand. The trends suggest that disorder-order transitions between binding site residues and distant residues are common and potentially a selected property of many proteins (*Kim et al., 2017*). Specifically, our data suggests that some of the entropy lost by the rigidification incurred by binding site residues upon ligand binding can be compensated with an increase in disorder in distant residues. This finding generalizes that the phenomenon has been observed in single-protein analyses with NMR and MD simulation (*Wang et al., 2019*; *Gohlke et al., 2004*; *Moorman et al., 2012*). Both theoretical and experimental analyses suggest that the relationship between local packing optimization and small voids that permit alternative conformations will be key to predictably mapping this relationship (*Caro et al., 2021*; *Bromberg and Dill, 1994*). Using temperature or pressure as perturbations during X-ray data collection can help to further map the connection between packing 'quality' and side-chain conformational heterogeneity in greater detail (*Caro et al., 2021*). While NMR order parameter studies only take into account movement that is shorter than the tumbling time for the protein (*Hoffmann et al., 2021*; *Gagné et al., 1998*), our results are insensitive to timescale. In addition, it is quite likely that our use of cryo-cooled structures causes an underestimate of the heterogeneity occurring in these datasets (*Fraser et al., 2011*), and may potentially bias our results by locking in certain populations of the protein ensemble. This effect may particularly impact areas of the protein and surrounding solvent that go from a preorganized, low-energy state to a more dynamic state as observed in galectin-3 and Barnase (*Diehl et al., 2010*; *Caro et al., 2021*). This study can also serve as a template to investigate other perturbations, including mutations, pressure, or temperature.

We observed a complex interplay between conformational change and dynamics in our analysis of 13 inhibitor-bound datasets of the kinase CDK2, in the same crystal form and space group. The ability to explore one protein with multiple ligands highlights the utility of crystal systems amenable to isomorphous soaking or co-crystallization (*Steuber et al., 2006*). We identified differences in conformational heterogeneity between type I and type II inhibitors that can be classified along with well-known changes, such as differences in the DFG motif. Tuning distal site dynamics may be a viable strategy for modulating the affinity of kinase inhibitors and affects the pattern of protein-protein interactions on distal surfaces, which is of critical importance in CDK inhibitor development (*Jhaveri et al., 2021*; *Wood and Endicott, 2018*).

We note that our work is not sensitive to many facets of the complex changes associated with ligand binding (*Mobley and Dill, 2009*). Our stringent resolution matching criterion may also render us blind to the most severe effects on conformational heterogeneity, whereby ligand binding causes a more widespread change, leading to a loss or gain of diffraction power. In addition, water molecules play an important role in ligand binding, both in the release of ordered water molecules contributing to binding through entropy and in forming specific interactions (*Breiten et al., 2013*; *Verteramo et al., 2019*). Additionally, ligand conformational heterogeneity has been highlighted by several recent studies (*van Zundert et al., 2018*; *Jain et al., 2020*; *Caldararu et al., 2021*). Another caveat in our analysis is the limitations of qFit modeling for modeling extensive backbone heterogeneity into weak electron density. Ensemble modeling methods, which leverage molecular dynamics for sampling and use a different model representation, may be more appropriate for examining these systems (*Burnley et al., 2012*; *Eshun-Wilson et al., 2019*). Future work, integrating the conformational heterogeneity of the protein, ligand, and water molecules, will create better predictions and explanations of the energetics of binding. In addition, this would allow us to interpret the impact of specific interactions and alterations on both the entropy and enthalpy of all components of the system.

Our study, as well as previous NMR studies (*Frederick et al., 2007*; *Caro et al., 2017*), only leverages a limited set of side-chain dihedral angles. However, comparisons with molecular dynamics simulations suggest that small sets of side-chain dihedrals alone may be representative of the overall changes in dynamics of the system (*Wand and Sharp, 2018*; *Kasinath et al., 2013*; *Chatfield and Wong, 2000*). What is the thermodynamic impact of restricting side-chain conformational

heterogeneity? Protein folding studies and theory indicate that restricting the rotamer of even a single side chain can incur an entropic penalty of binding of ~0.5 kcal/mol (*Doig and Sternberg, 1995*). While we observe many such restrictions in binding sites due to specific interactions with ligands, our data point to corresponding changes away from the binding site that help balance this cost. Overall, the median increase in rigidity we observe in binding site residues (0.03 order parameter increase) would create an energetic penalty of approximately ~0.1–0.5 kcal/mol, based off of the entropy meter calculated in *Caro et al., 2017* and *Caro et al., 2021*, with outliers having even larger thermodynamic consequences. Given the constraints of maintaining a folded conformational ensemble upon ligand binding, it is likely that ligand binding generally acts to restrict degrees of freedom locally and that protein topological constraints favor increased motion in distal regions (*Bromberg and Dill, 1994*). This overall effect likely manifests because optimizing affinity is desirable for medicinal chemistry and the selective pressures experienced by many proteins. Such optimization is insensitive as to whether the free energy is driven enthalpically or entropically. However, given the attention paid to designing and optimizing enthalpic interactions, there is likely unleveraged potential in optimizing the entropic component as well. As more sophisticated models of conformational heterogeneity are created and validated (*Rosenbaum et al., 2021*), the strategy of rationally tuning conformational heterogeneity to improve binding affinity may be an attainable design strategy.

## Materials and methods
### Dataset
Our dataset was compiled using a snapshot of the PDB (*Berman, 2002*) in September 2019, containing 156,187 structures. We then selected structures that had a resolution better or equal to 2 Å (n = 64,557). We also excluded any structure that contained nucleic acids (n = 2280) or covalently bound ligands (n = 1030). We identified holo structures (n = 30,530), defined as those that contained at least one ligand, defined as any HETATM residue with 10 or more heavy atoms, excluding common crystallographic additives.

To create apo/holo pairs, we took each holo structure and compared it to each potential apo structure (n = 30,717), defined as structures without a ligand bound. A pair was defined according to the following criteria:

- Same space group.
- Exact sequence or exact sequence after removing the first or last five base pairs of either structure.
- A resolution difference between the two structures less than 0.1 Å.
- Dimensions of unit cells do not differ by more than 1 Å
- Angles of the unit cells do not differ by more than 1°.

This gave us 15,214 pairs. We then subsetted this list down to provide only one apo structure per holo structure, based on the smallest resolution difference. This produced a final pair set of 1205 with 1143 unique structures (*Supplementary file 2*).

We also created a pairset with 458 unique apo/apo pairs using the same criteria as the apo/holo pairset (*Supplementary file 3*).

### Refinement
We re-refined all structures using phenix.refine (https://www.phenix-online.org/documentation/reference/refinement.html). This was done using Phenix version 1.17.1–3660. We performed anisotropic refinement on all pairs where both PDBs had a resolution better than 1.5 Å. All other refinement was run isotropically. Refinement used the following parameters:

- Refine strategy: individual sites + individual adp + occupancies.
- Number of macro cycles: eight.
- NQH flips: true.
- Optimize xyz weight: true.
- Optimize adp weight: true.
- Hydrogen refine: riding.

We removed 102 structures because of incompatibility with our re-refinement pipeline due to breaks in the protein chain or ligand incompatibility. We removed 88 structures where the R-free increased by >2.5% compared to the value reported in the PDB header (*Supplementary file 1*; *Figure 1—figure supplement 2D*).

## Running qFit

qFit-3.0 (*Riley et al., 2021a*; version 3.2.0) was run using a composite omit map and the re-refined structure on the default parameters (https://github.com/ExcitedStates/qfit-3.0/). We ran qFit on Amazon Web Services (AWS). We used an autoscaling cluster of images controlled by the scheduler via ParallelCluser. Please see the qFit GitHub for a script that will install qFit on AWS's default OS image using Conda to install its dependencies.

After qFit, we reran refinement as suggested by qFit-3.0. Briefly, this involves three rounds of refinement. The first refines coordinates only, the second goes through a cyclical round of refinement until the majority of low-occupancy conformers are removed, and the third refinement polishes the structure, including hydrogen. The script used for post-qFit refinement can be found at https://github.com/ExcitedStates/qfit-3.0/blob/master/scripts/post/qfit_final_refine_xray.sh. We removed 100 structures because of incompatibility with refinement after qFit rebuilding.

## Quality control

From our original dataset (n = 1205 pairs), we removed 28 apo structures that had a crystallographic additive or amino acid in the binding site that partially overlaid with the holo structure. We set a minimum ligand occupancy threshold of 0.15, which did not remove any pairs from our dataset. Chains were renamed according to the difference in distance between the two chains. We also renumbered each chain based on the apo structure. We then superimposed the two structures using the PyMOL align function. We measured the alpha carbon RMSD between the two structures as well as the difference in just binding site residues. Structures were removed if the mean RMSD of the entire structure was greater than 1 Å or if the mean RMSD in the binding site residues was greater than 0.5 Å. We removed two pairs based on these RMSD criteria.

We also assessed the difference in R-free values for each refinement step (before/after qFit). If the post-refinement R-free value was 2.5% larger than the pre-refinement R-value, then the structure was removed (n = 85,77 structures removed; *Figure 1—figure supplement 2A and B*). Additionally, we compared the final R-free values between apo and holo pairs, removing pairs with R-free values with more than a 5% difference (n = 16 pairs removed; *Figure 1—figure supplement 2C*). We ran the clashscore function out of MolProbity (*Williams et al., 2018*) to identify severe clashes in our dataset. We removed any structures with a clashscore greater than 15, removing 52 structures. After filtering out pairs that failed our quality checks, our dataset contained 743 matched apo/holo pairs.

## Alternative conformations

Side chains were considered alternative conformers if there was at least one atom that was modeled with an alternative conformer. Our re-refinement procedure changes the occupancy, coordinates, and B-factors of these conformations, but it does not add or delete conformations.

## B-factors

B-factors were assessed on a residue basis by averaging the B-factors of all heavy atoms for each residue. For residues with multiple conformations, we took the mean B-factor for all heavy atoms in each side chain, weighted by occupancy. For structures modeled anisotropically, we used the isotropic equivalent B-factor from Phenix.

## Root mean squared fluctuation

RMSF was chosen over root mean squared deviation as many alternative conformers were predicted to have the same occupancy, thus making it difficult to define which was the main conformer. RMSF was measured for each residue based on all side-chain heavy atoms. RMSF finds the geometric center of each atom in all alternative conformers. It then takes the distance between the geometric mean of each conformer's side-chain heavy atoms and the overall geometric center. It then takes the squared mean of all of those distances, weighted by occupancy.

## Order parameters

Order parameters were measured for each residue (except proline and glycine) by taking into account both the angle of alternative conformers (s2angle), by measuring the chi1 angle, and the B-factors of alpha or beta carbons along with an attached hydrogen(s2ortho) (*Fenwick et al., 2014*). To account for differences in B-factors as resolution changes, we investigated the correlation between order parameters in 32 apo lysozyme structures ranging in resolution from 1.1 to 2 Å. We optimized the s2ortho parameter by looking for the normalization that would provide a slope closest to 1 and have the smallest root mean squared error (*Figure 3—figure supplement 1A and B*, *Supplementary file 4*). We normalized the s2ortho portion using the following equation:

$$s2ortho_{normalized} = s2ortho * Bfactor_{alpha\ carbon}/10\ (resolution)$$

The final order parameter reported in the article is

$$s2calc\ =\ s2ortho_{normalized} * s2ang$$

## Rotamer analysis

Rotamers were determined using phenix.rotalyze (*Williams et al., 2018*) with manually relaxing the outlier criteria to 0.1%. Each alternative conformation has its own rotamer state. Rotamers were compared on a residue-by-residue basis between the holo and apo, taking into account each rotamer state for each alternative conformation. Residues were classified as 'no change' if rotamers matched across the apo and holo residue, 'distinct' if the matched residue shared no rotamer assignments. Residues were classified as 'remodeled-holo loss' if distinct, additional rotameric conformations were populated in the apo residue only, and 'remodeled-holo gain' if distinct, additional rotameric conformations were populated in the holo residue only.

## Hydrogen bond analysis

To assess for the changes in hydrogen bonding across all pairs in our study, we applied HBplus (*McDonald and Thornton, 1994*) to every multiconfomer structure. HBplus identifies hydrogen bonds when the distance between the hydrogen and acceptor is less than 3.2 Å, with a maximum angle of 90°. Since HBplus, nor any other software program we could identify, looks at hydrogen bonds in reference to alternative conformers, we split up each multiconformer PDB by alternative conformation. For example, the altA PDB contained all atoms that had an alternative conformer A as well as all atoms with no alternative conformation.

We then examined all of the hydrogen bonds for each PDB in binding site residues. We only considered hydrogen bonds between side chains or between side chains and the main chain. Hydrogen bonds were weighted based on the lowest occupancy of the acceptor or donor atom. We then controlled for the number of residues in the binding site.

## Solvent-exposed surface area

We calculated the relative accessible surface area (RASA) using Define Secondary Structure of Proteins (DSSP) (*Kabsch and Sander, 1983*) with the *Tien et al., 2013* definition of Max accessible surface area (MaxASA). Residues with a RASA of ≥20% were considered solvent exposed (*Wu et al., 2017*).

## Ligand analysis

We obtained the ligand properties using RDkit (version 3/2/2021) by importing SDF files of each ligand in our dataset. To account for the multiple hypothesis testing, we applied a Bonferroni correction, with an alpha of 0.05, as we were testing 10 hypotheses, leaving us with a corrected significance value of 0.005.

Occupancy of the ligands was taken directly from the PDB file and corresponds to the ligand occupancy from the deposited structure. Ligand B-factors were normalized using the mean alpha carbon B-factor of all residues in the structure.

If there were multiple ligands of interest in a structure, we looked at the properties of the ligand and surrounding protein residues in chain A or in the lowest alphabetical chain.

## Protein-type analysis

Protein names and enzyme names were extracted from *UniProt Consortium, 2015*. Names and properties were connected using PDB IDs.

## Ringer analysis

Indvidual residues in the CDK2 structures were run through Ringer using mmtbx.ringer. Outputs from the csv file were then plotted using Matplotlib.

## Statistics

Paired Wilcoxon test was used for all matched properties (comparing holo vs. apo matched residues or structures). Individual Mann–Whitney *U* test was used for all non-match properties, including ligand properties. Two-sided *t*-test was used to compare the significance of the slopes.

## Code

Code can be found in the following repositories:

- Dataset selection: https://github.com/fraser-lab/Apo_Holo_Analysis.
- Refinement/qFit pipeline: https://github.com/fraser-lab/Apo_Holo_Analysis.
- Analysis/figures: https://github.com/fraser-lab/Apo_Holo_Analysis; *Wankowicz et al., 2022* copy archived at swh:1:rev:c92e50c121624b2e4ce440586225b8d7c48dfe38.
- qFit: https://github.com/ExcitedStates/qfit-3.0; *Riley et al., 2021b*.

# Acknowledgements

We thank James Holton, Tony Capra, and Dan Herschlag for helpful comments and suggestions. JSF was supported by NIH GM123159 and GM124149, and a Sanghvi-Agarwal Innovation Award. SAW was supported by NSF GRFP 2034836.

# Additional information

### Competing interests

Saulo H de Oliveira, Henry van den Bedem: is an employee of Atomwise Inc. James S Fraser: has equity, has received consulting fees, and has sponsored research agreements with Relay Therapeutics. The other authors declare that no competing interests exist.

### Funding

| Funder | Grant reference number | Author |
|---|---|---|
| National Science Foundation | GRFP 2034836 | Stephanie A Wankowicz |
| National Institutes of Health | GM123159 | James S Fraser |
| National Institutes of Health | GM124149 | James S Fraser |

The funders had no role in study design, data collection and interpretation, or the decision to submit the work for publication.

### Author contributions

Stephanie A Wankowicz, Conceptualization, Data curation, Formal analysis, Investigation, Methodology, Software, Validation, Visualization, Writing – original draft, Writing – review and editing; Saulo H de Oliveira, Conceptualization, Investigation, Methodology, Software, Writing – review and editing; Daniel W Hogan, Investigation, Resources, Software, Writing – review and editing; Henry van den Bedem, Conceptualization, Project administration, Writing – review and editing; James S Fraser,

Conceptualization, Funding acquisition, Project administration, Resources, Supervision, Writing – original draft, Writing – review and editing

**Author ORCIDs**
Stephanie A Wankowicz http://orcid.org/0000-0002-4225-7459
James S Fraser http://orcid.org/0000-0002-5080-2859

**Decision letter and Author response**
Decision letter https://doi.org/10.7554/eLife.74114.sa1
Author response https://doi.org/10.7554/eLife.74114.sa2

## Additional files

**Supplementary files**
- Supplementary file 1. Crystallographic additives.
- Supplementary file 2. Apo/holo pairs.
- Supplementary file 3. Apo/apo pairs.
- Supplementary file 4. Lysozyme normalization IDs.
- Supplementary file 5. Nuclear magnetic resonance (NMR) vs. X-ray order parameters in lysozyme.
- Transparent reporting form

**Data availability**
Refined models are available here: https://doi.org/10.5281/zenodo.6474333. Code can be found in the following repositories: https://github.com/fraser-lab/Apo_Holo_Analysis (copy archived at swh:1:rev:c92e50c121624b2e4ce440586225b8d7c48dfe38) and https://github.com/ExcitedStates/qfit-3.0.

The following dataset was generated:

| Author(s) | Year | Dataset title | Dataset URL | Database and Identifier |
|---|---|---|---|---|
| Wankowicz SA, de Oliveira SHP, Hogan DW, van den Bedem H, Fraser JS | 2021 | Ligand binding remodels protein side chain conformational heterogeneity | https://doi.org/10.5281/zenodo.6474333 | Zenodo, 10.5281/zenodo.6474333 |

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
