## [Editor Report]

This work attempts to extract information about protein thermodynamics from X-ray crystallography data, which is a challenging problem. The heterogeneous pattern of order parameter changes in response to ligand binding implies that the approach is identifying new information. This work offers insights into ligand binding affinity and specificity mechanisms, suggesting that distal (allosteric) perturbations represent a possible avenue to modulate protein function.

---

## [Decision Letter]

**Decision letter after peer review:**

Thank you for submitting your article "Ligand binding remodels protein side chain conformational heterogeneity" for consideration by *eLife*. Your article has been reviewed by 3 peer reviewers, and the evaluation has been overseen by a Reviewing Editor and Volker Dötsch as the Senior Editor. The reviewers have opted to remain anonymous.

Essential revisions:

1. A discussion of the forces that dominate ligand binding affinities in the context of the analysis of these matching pairs would be desirable. In particular, H-bond networks in the apo and holo forms inferred from the structural coordinate should be compared and included in the structural characterizations of rearrangements due to ligand binding. Moreover, NMR and neutron diffraction (doi: 10.1021/acs.biochem.5b00387; doi: 10.1126/sciadv.aav0482) suggest a global rearrangement of the H-bond network follows ligand binding, at least in the case of kinases. Please analyze and discuss H-bond networks.

2. What is the ligand affinity distribution and is there a correlation of affinity with the effects observed? We realize that ligand affinity is not easily accessible in the literature and may require collection "by hand" and could be an unreasonable task for the entire database of structures. However, a reasonable subset of complexes could be examined from this point of view. For example, for affinities for the Galectin system cover a wide range.

3. The "no net contribution" claim for conformational entropy is at odds with Caro et al. 2017. Possible reasons are incomplete representation of solution behavior at cryogenic temperatures and the inability of the approach used here to monitor the three "classes" of rotamer motion sensed by NMR and MD. Please comment and discuss.

4. The choice of defining the active site as all heavy atoms within 5 Å of a ligand heavy atom seems reasonable, but given the importance of this definition to the result, some exploration {plus minus} 5 Å seems warranted. For a given active site distance definition, what fraction of the residues included are considered direct contacts with substrate? How does this influence rigidity? Presumably, a plot of distance threshold vs. number of ligand contacts made over the dataset could help justify this. A related issue concerns the use of different non-binding site definitions. Several terms are mentioned-distant-binding, and far-binding-and they seem to be used interchangeably in the text and in figure labels/captions. No criteria are explicitly mentioned in one case, so this is presumably all residues outside of 5 Å (line 318, Supplementary Figure 7B), while a 10 Å, <20% solvent exposure criteria is invoked later (line 323, Figure 3C). It would be helpful to clean up this terminology and expand on the latter criteria. To what degree does the effect observed depend on this distance cutoff? It appears as though Supplementary Figure 7C and Figure 3C are quite similar-how sensitive is the difference in slopes between the two to this cutoff? Concerning the sensitivity of the linear fits, are the slopes and correlations sensitive to the exclusion of randomly chosen holo-apo pairs? Additionally, are any proteins small or oddly shaped enough to either be effectively excluded by this criteria and, more generally, how does protein size correlate with the observed effect? Please investigate the sensitivity of the conclusions to the particular selections and criteria.

5. The discussion (186-197) regarding analysis of the redistribution of rotamers and the later connection to entropy somewhat glosses over the issues of resolving not only major alternate rotamers (the focus here) but also motion within a rotamer well with or without occasional crossing into another rotamer carries significant entropy and that changes between these latter classes can contribute significant to binding entropy (Frederick et al. 2007; Caro et al. 2017; Rajeshwar et al. J. Phys. Chem. B 2021, 125, 9641). Please comment and discuss.

6. The observation (269-275) that the net change in "conformational heterogeneity" is near zero is in conflict to a room temperature observation in solution that changes in fast motion (conformational entropy) can strongly oppose, be negligible, or strongly favor the thermodynamics of ligand binding (Caro et al. 2017). Please comment and discuss.

7. The section describing conformational change and heterogeneity in CDK2 is appreciated. However, the authors allude to agreement with the work of Kim et al. 2017 on PKA but show no quantitative comparison with it (or other relatively recent work on PKA). More generally, the authors note that the results of this work are consistent with studies in various specific systems, but no direct comparisons are made. For example, Moorman et al. 2012 cited by the authors described the rigidification of lysozyme in the presence of the ligand based on NMR experiments. Figure 6 from that paper is directly analogous to the work presented in this manuscript. According to Supplementary Figure 4C of this manuscript, lysozyme is the second most represented protein in the author's dataset, but no comparisons to the work of Moorman et al. (2014) who provided a fitted fall off distance dependence for the binding of substrate (inhibitor) binding to Ras Cdc42 and also showed the presence of "sidedness" and vectoral (channeled) transmission of dynamical perturbation. Please comment and discuss.

8. While very detailed comparisons of order parameters from NMR experiments and crystallography data have been made previously in Fenwick et al. 2013 and are perhaps outside of the scope of this paper, some demonstration of agreement between the new results presented here and existing work would be much appreciated. More generally, a number of previous studies have explored the idea of propagated structural change in response to not only ligand binding but other perturbations such as pressure, temperature, saturation mutagenesis, etc.; are the spatial patterns reported here consistent with results from more orthogonal approaches in any of the proteins in the dataset? Please comment and discuss.

9. On the interface between ligand and protein, the statement (389-391) "an intuitive general picture emerges where more specific interactions, such as hydrogen bonds, are correlated with more rigid binding site residues, whereas the more non-specific interactions are correlated with more flexible binding site residues." needs some qualification. Why is a VDW contact considered "non-specific"? Do the authors imply that buried polar interactions are more stable/rigid than simple non-polar VDW interactions? Complementarity of motion in small molecule ligands is difficult to probe, as noted, but has been examined in protein-protein interactions, particularly in the calmodulin complexes (Marlow et al. Nat Chem Biol 6, 352). Please comment and discuss.

[Editors' note: further revisions were suggested prior to acceptance, as described below.]

Thank you for resubmitting your work entitled "Ligand binding remodels protein side chain conformational heterogeneity" for further consideration by *eLife*. Your revised article has been evaluated by Volker Dötsch (Senior Editor) and a Reviewing Editor.

The manuscript has been much improved but there are some remaining issues that need to be addressed, as outlined below:

The authors explore the effect of different active site definitions on the change in binding site order parameters upon ligand binding. While the authors originally reported this response at a fixed cutoff of 5 Å, they have included new results exploring the change at integer-valued distances from 2-10 Å. This analysis reveals a trend perhaps consistent with expectation, where the median change in order parameter starts high with very tight active site definitions and falls off with increasingly large active sites. In the absence of any surprising features on this curve, the 5 Å cutoff seems reasonable. Please include this figure in the final manuscript (perhaps as a supplemental figure).

The authors evaluated the robustness of the linear regressions between order parameters of non-solvent exposed residues versus average order parameters of binding site residues for holo-apo and apo-apo pairs via bootstrap analysis. F3S2E clearly shows that the mean slopes are robust to exclusion of data in either case. This is a compelling result that greatly improves the paper. Perhaps this result should be emphasized further in the paper.

The authors demonstrated that the majority of the proteins under consideration appear to have an active site composed of a fraction of the total residues less than half of the total number of residues total, and that the number of outliers in this comparison (i.e., with more active site residues than distant residues) is small and thus unlikely to bias the result. The question "how does protein size correlate with the observed effect" is left unanswered and may be scientifically interesting, but perhaps the analysis performed is sufficient to demonstrate that there are at least no pathologies associated with active site definitions. Please comment.

Regarding the comparison to the PKA data from Kim et al. 2017 in point 7, the authors note that a direct comparison is challenging, due to the specifics of the different experiments. The authors clarify that they mean that the overall nature of the changes upon ligand binding are similar (i.e., rigidification throughout the protein). Please clarify the word "pattern" here, as it suggests something more residue-specific, although the overall statement seems fair now.

The addition of F3S2F in point 7 is welcome, although side-by-side comparison with Moorman et al. 2012 F6 is challenging to interpret. Moorman et al. identify a specific subset of contiguous residues in lysozyme which rigidify upon ligand binding, and several that relax. Do the authors intend to imply that the same residues rigidify with ligand binding as in the previous study, or are they saying that there is some more general spatial pattern that is similar? Is there no quantitative comparison that can be made? Explicitly, do the ω-class residues that show collective rigidification upon ligand binding (A11, V29, I55, L83, A90, V92, M105, A107) also rigidify based on the analysis presented here? Do the peripheral residues that relax upon ligand binding (L25, T69) also relax in the analysis presented here? It is very difficult to make such an assessment by staring at F3S2F. There may not be a matching crystal structure for chitotriose, but one might expect some overlap in the residues which rigidify or relax the most for other similar ligands. Furthermore, the question about a similar comparison with Moorman et al. 2014 for Ras Cdc42 still needs to be addressed. Please comment and discuss.

Please provide some form of quantitative comparison between the Moorman et al. 2012 results and the results here for lysozyme.

---

## [Author Response]

Essential revisions:1. A discussion of the forces that dominate ligand binding affinities in the context of the analysis of these matching pairs would be desirable. In particular, H-bond networks in the apo and holo forms inferred from the structural coordinate should be compared and included in the structural characterizations of rearrangements due to ligand binding. Moreover, NMR and neutron diffraction (doi: 10.1021/acs.biochem.5b00387; doi: 10.1126/sciadv.aav0482) suggest a global rearrangement of the H-bond network follows ligand binding, at least in the case of kinases. Please analyze and discuss H-bond networks.

To assess for the changes in hydrogen bonding across all pairs in our study, we applied HBplus (McDonald, I. K., and Thornton, J. M. (1994). Satisfying hydrogen bonding potential in proteins. *Journal of molecular biology*, *238*(5), 777-793.) to every multiconfomer structure. HBplus identifies hydrogen bonds when the distance between the hydrogen and acceptor are less than 2.5 Å, with a maximum angle of 90 degrees. Since HBplus, nor any other software program we could identify, does not look at hydrogen bonds in reference to alternative conformers, we split up each multiconformer PDB by alternative conformation into separate single conformer PDB files. For example, the alt A PDB contained all atoms that had an alternative conformer A as well as all atoms with no alternative conformation, the alt B PDB contained all atoms contained all atoms that had an alternative conformer B as well as all atoms with no alternative conformations, etc.

We then examined all of the hydrogen bonds for each PDB across the entire PDB as well as in just binding site residues. We only considered hydrogen bonds between side chains or between side chains and the main chain. Hydrogen bonds were weighted based on the lowest occupancy of the acceptor or donor atom. We then controlled for the number of residues in the binding site. Figure 3- Supplement Figure 3 shows the distribution of the difference of hydrogen bonds between holo and apo structures (A). We observed a net gain of 0.06 hydrogen bonds per residue in holo binding sites, with ~10% of structures gaining one full hydrogen bond in the holo structure. This is indicative of a trend towards more conformationally stable binding sites in holo structures.

As an example of an outlier pair, we looked at (Holo: 2PLL, Apo: 2PHA) that had 2.2 more hydrogen bonds (averaged by occupancy) in the holo compared to the apo. We identified the three residue pairs that had hydrogen bonds in the holo that were not in the apo structure.

All of these were due to some of the conformations of one or both of the side chains having a dramatically different confirmation in the holo (purple) compared to the apo (green). In the first pair of residues (B), we observe W118 gains a hydrogen bond with H122 due to a conformation that is not present in the apo ensemble. In the second pair of residues (C), K64 in the apo structure does not make any hydrogen bonds with S133, but the last chi angle of K64 is more constrained in the holo ensemble. In the third pair of residues (D), alt A and B in H97 of the apo structure have a much different conformation from H97 in the holo structure. We have included the following paragraph in the paper (line 451-460) and included Figure 3- Supplement Figure 3.

The two kinase papers mentioned by the reviewers use HDX and Neutron crystallography to show that there are differences in the strengths of various hydrogen bonds. The largest conclusions from that are that: (1) the backbone hydrogen bonds in the DFG region change depending on activation status; (2) that outside the DFG, most H-bonds get stronger in the “catalytic” complex relative to the “apo” complex. Direct comparison to our data is complicated: PKA is quite diverged from CDK-2 and we are examining inhibited complexes, not models of the “catalytic” complex. Our analysis of hydrogen bonds in CDK-2 reveals a complex pattern. Many residues lose hydrogen bonds, particularly in loop regions including the activation loop (A) and there is a difference in hydrogen bonds gained that depends greatly on the inhibitor.

We have added the following to the text of the paper in lines (635-637 and 643-645):

‘We also observed that many of these residues had sidechain to sidechain hydrogen bonds that were lost upon ligand binding (Figure 5—figure supplement 2A).’

‘We also observed that hydrogen bonds gained in the holo structure are inhibitor specific (Figure 5—figure supplement 2B/C).’

We have also added Figure 5—figure supplement 2.

2. What is the ligand affinity distribution and is there a correlation of affinity with the effects observed? We realize that ligand affinity is not easily accessible in the literature and may require collection "by hand" and could be an unreasonable task for the entire database of structures. However, a reasonable subset of complexes could be examined from this point of view. For example, for affinities for the Galectin system cover a wide range.

To examine the binding affinities of the ligands in our dataset, we connected the PDB codes to ChEMBL (https://www.ebi.ac.uk/chembl/) to obtain binding data. From here, we only examined Ki as this was the most abundant type of binding information we had. We had 105 ligand-protein pairs with binding data. The range of affinities varied greatly from 0.02 nM to 486000 nM. We correlated the log(ki) values with both the binding site order parameters for the holo protein as well as the difference in binding site order parameters between the apo and holo protein. We did not observe any correlation here (see Author response image 1).

**Author response image 1. sa2fig1:** The relationship between order parameters and binding affinity. (A) The difference in average binding site order parameters (holo-apo) are not correlated with the inhibitory constant (Ki) of binding. (B) The average order parameter of the holo binding site residues are not correlated with the inhibitory constant (Ki) of binding.

We also examined binding affinities of CDK2 and Trypsin, of which we were able to obtain binding data from six and 10 protein-ligand pairs, respectively. The binding data from each of these analyses were obtained from the original paper and used the same concentration of natural ligand to determine the ki. Again, we were not able to detect any correlation between our metrics of conformational heterogeneity and the binding affinity. These results are not unexpected given the diversity of ligands we have in our dataset – and in our view – do not detract from the idea that altering protein conformational heterogeneity in response to changes within a ligand series is expected to alter binding thermodynamics. We would expect the residual order parameter to be most correlated in the context of a congeneric series of ligands, not the diverse ligands that are reported in the literature.

**Author response image 2. sa2fig2:** We looked at the correlation between order parameters and the inhibitor constant (Ki) of binding for the five CDK2 protein-inhibitor complexes we had data for. We did not observe any trends in this data. (A) The difference in average binding site order parameters (holo-apo) are not correlated with the inhibitory constant (Ki) of binding. (B) The average order parameter of the holo binding site residues are not correlated with the inhibitory constant (Ki) of binding. (C) The relationship between the residual order parameter (Distant Residues – Binding Site Residues).

**Author response image 3. sa2fig3:** We looked at the correlation between order parameters and the inhibitor constant (Ki) of binding for the 10 trypsin protein-inhibitor complexes we had data for. We did not observe any trends in this data. (A) The difference in average binding site order parameters (holo-apo) are not correlated with the inhibitory constant (Ki) of binding. (B) The average order parameter of the holo binding site residues are not correlated with the inhibitory constant (Ki) of binding. (C) The relationship between the residual order parameter (Distant Residues – Binding Site Residues).

3. The "no net contribution" claim for conformational entropy is at odds with Caro et al. 2017. Possible reasons are incomplete representation of solution behavior at cryogenic temperatures and the inability of the approach used here to monitor the three "classes" of rotamer motion sensed by NMR and MD. Please comment and discuss.

We concluded that on average, across our 743 pairs, there was no net trend in whether conformational entropy increased or decreased upon ligand binding. However, the spread away from the average trend for individual proteins suggests that there is a large distribution of changes in conformational entropy for individual proteins which is in line with Caro et al., which states ‘Conformational entropy can highly disfavor, have no effect on, or strongly favor association and it is often a large determinant of the thermodynamics of binding’.

We have updated the text in the discussion to clarify this (lines 662-664).

‘We observed that individual proteins greatly varied in amount and direction of change of conformational heterogeneity, as observed in previous studies (Caro et al., 2017).’

4. The choice of defining the active site as all heavy atoms within 5 Å of a ligand heavy atom seems reasonable, but given the importance of this definition to the result, some exploration {plus minus} 5 Å seems warranted. For a given active site distance definition, what fraction of the residues included are considered direct contacts with substrate? How does this influence rigidity? Presumably, a plot of distance threshold vs. number of ligand contacts made over the dataset could help justify this.

Note, we decided on the 5Å cut off before exploring our results. So, while this analysis shows an even stronger trend at other distances, we will keep the focus of the analysis on the 5Å cut off in the text.

To explore the difference in distance threshold, we looked at the difference in binding site order parameters (holo-apo) at nine different cutoffs (2-10Å). We see a general trend of the smaller the distance threshold, the more pronounced the increased in order parameters from apo to holo (represented by a positive value). See Figure 3—figure supplement 2.

A related issue concerns the use of different non-binding site definitions. Several terms are mentioned-distant-binding, and far-binding-and they seem to be used interchangeably in the text and in figure labels/captions.

We have gone through the paper and revised the language related to the non-binding site definitions to make sure they are uniformly labeled with “distant residues” versus “binding site residues”.

No criteria are explicitly mentioned in one case, so this is presumably all residues outside of 5 Å (line 318, Supplementary Figure 7B), while a 10 Å, <20% solvent exposure criteria is invoked later (line 323, Figure 3C). It would be helpful to clean up this terminology and expand on the latter criteria. To what degree does the effect observed depend on this distance cutoff?

We also included the following sentence in the Results section clarifying which residues were included in Figure 3- Supplement Figure 2 (line 409-410).

‘Therefore, we explored the relationship between binding site residues and distant residues, defined as those more than 10Å away from any heavy atom in the ligand.’

It appears as though Supplementary Figure 7C and Figure 3C are quite similar-how sensitive is the difference in slopes between the two to this cutoff? Concerning the sensitivity of the linear fits, are the slopes and correlations sensitive to the exclusion of randomly chosen holo-apo pairs?

Author response image 4 shows the two plots on top of each other demonstrating that the slope of the holo versus apo is steeper (-0.44 versus -0.28). This emphasizes our conclusion that the ligand perturbs a naturally occurring phenomenon seen in apo proteins alone.

**Author response image 4. sa2fig4:** We plotted the distribution of the average difference in order parameters in binding site residues (holo-apo/apo-apo) versus the residual order parameter distant residues versus binding site residue to highlight the slight difference in their slopes (-0.28, apo/apo versus -0.44 apo/holo).

Additionally, we have calculated the slope of the holo versus apo line in a bootstrap manner. Here we have calculated the slope 10,000 times based on a random set of 743 datapoints, allowing for replacement. We also did the same analysis with the apo-apo dataset. Figure 3- Supplement Figure 3 shows a histogram of the holo-apo bootstrapped slope values versus the apo-apo bootstrapped slope values.

The average holo-apo slope over the 10,000 bootstrapped samples was -0.44, with a standard deviation of 0.029. The average apo-apo slope over the 10,000 bootstrapped samples was -0.29, with a standard deviation of 0.072. While there is some overlap in the bootstrapped values, the apo-apo slope mean value is more than two standard deviations away from the holo-apo mean slope value. Comparing this using a z-test, the z-value was -191.26 with a p-value of 0.0.

Additionally, are any proteins small or oddly shaped enough to either be effectively excluded by this criteria and, more generally, how does protein size correlate with the observed effect? Please investigate the sensitivity of the conclusions to the particular selections and criteria.

To assess for the differences in the number of residues considered in the binding site versus in the distant, non-solvent exposed, we calculated the ratio of the binding site residues versus the distant non-solvent exposed residues. The majority of PDBs had more residues in the distant non-solvent (represented as those with a ratio under 1 in the Author response image 5). There were a few outliers with very few distant non-solvent exposed residues. These represented both small proteins and PDBs with the ligand buried deep into the center of the protein (example: 2RCT, ligand in salmon, Autor response image 5). We then checked the δ OP in the outliers (>2 ratio between binding/distant residue), which are colored in red. They all fall within the distribution we observed from other samples.

**Author response image 5. sa2fig5:** (A)The distribution of the ratio of binding versus distant residues in all structures in our dataset (B) The distribution of the relationship between the difference in binding site order parameters and distant order parameters. The outliers (>2 residue ratio are highlighted in red). (C) An example of a structure (PDB ID: 2RCT) with very few distant residues, one of the outliers in our dataset.

5. The discussion (186-197) regarding analysis of the redistribution of rotamers and the later connection to entropy somewhat glosses over the issues of resolving not only major alternate rotamers (the focus here) but also motion within a rotamer well with or without occasional crossing into another rotamer carries significant entropy and that changes between these latter classes can contribute significant to binding entropy (Frederick et al. 2007; Caro et al. 2017; Rajeshwar et al. J. Phys. Chem. B 2021, 125, 9641). Please comment and discuss.

We agree that there is significant entropy within rotamer wells, which we emphasize in the discussion of B-factors and the incorporation of B-factors in the crystallographic order parameter calculation. Since the bulk of the results focus on order parameters which combine the motion between and within rotamer wells, we added the following section to the introduction of order parameters to further drive this message home (lines 316-317).

‘Order parameters allow us to capture the conformational entropy both within and between side chain rotamer wells.’

6. The observation (269-275) that the net change in "conformational heterogeneity" is near zero is in conflict to a room temperature observation in solution that changes in fast motion (conformational entropy) can strongly oppose, be negligible, or strongly favor the thermodynamics of ligand binding (Caro et al. 2017). Please comment and discuss.

As discussed in point 3 above, we concluded that on average, across our 743 pairs, there was no trend in whether conformational entropy increased or decreased upon ligand binding for the entire dataset. However, across the pairs we observed a large distribution of changes in conformational entropy which is in line with Caro et al., which states ‘Conformational entropy can highly disfavor, have no effect on, or strongly favor association and it is often a large determinant of the thermodynamics of binding’.

We have updated the text in the discussion to clarify this (lines 662-664).

‘We observed that individual proteins greatly varied in amount and direction of change of conformational heterogeneity, as observed in previous studies (Caro et al., 2017).’

7. The section describing conformational change and heterogeneity in CDK2 is appreciated. However, the authors allude to agreement with the work of Kim et al. 2017 on PKA but show no quantitative comparison with it (or other relatively recent work on PKA). More generally, the authors note that the results of this work are consistent with studies in various specific systems, but no direct comparisons are made.

We wanted to make the connection between our findings of the dispersion and connectedness of changes observed in Kim et al., however, due to sequence differences and the fact that Kim et al. only measured certain residues, we are unable to compare these quantitatively. We have clarified that the consistency is due to the widespread nature of changes in conformational dynamics in kinases systems as a function of ligand state. We have amended the following sentence in lines 580-583.

‘This dispersed pattern is similar to the trend of rigidification that is observed by NMR in PKA upon substrate binding, suggesting that changes in conformational dynamics in kinases systems are structurally dispersed as a function of ligand state (Kim et al. 2017).’

For example, Moorman et al. 2012 cited by the authors described the rigidification of lysozyme in the presence of the ligand based on NMR experiments. Figure 6 from that paper is directly analogous to the work presented in this manuscript. According to Supplementary Figure 4C of this manuscript, lysozyme is the second most represented protein in the author's dataset, but no comparisons to the work of Moorman et al. (2014) who provided a fitted fall off distance dependence for the binding of substrate (inhibitor) binding to Ras Cdc42 and also showed the presence of "sidedness" and vectoral (channeled) transmission of dynamical perturbation. Please comment and discuss.

To compare the overall trend of differences in order parameters in a lysozyme bound to chitotriose (as observed in Moorman et al. 2012), we compared the difference in order parameters between holo and apo structure of 4XEN, which is bound to acetylchitotetraose. We observed a similar trend to Moorman et al. 2012, where there was a central core of residues that become more rigid upon ligand binding (blue).

8. While very detailed comparisons of order parameters from NMR experiments and crystallography data have been made previously in Fenwick et al. 2013 and are perhaps outside of the scope of this paper, some demonstration of agreement between the new results presented here and existing work would be much appreciated. More generally, a number of previous studies have explored the idea of propagated structural change in response to not only ligand binding but other perturbations such as pressure, temperature, saturation mutagenesis, etc.; are the spatial patterns reported here consistent with results from more orthogonal approaches in any of the proteins in the dataset? Please comment and discuss.

While we agree that these comparisons are needed, we believe they are beyond the scope of this paper. However, given the importance of this analysis, we have added to the discussion of how we would approach this (lines 686-687).

‘This study can also serve as a template to investigate other perturbations including mutations, pressure, or temperature.’

9. On the interface between ligand and protein, the statement (389-391) "an intuitive general picture emerges where more specific interactions, such as hydrogen bonds, are correlated with more rigid binding site residues, whereas the more non-specific interactions are correlated with more flexible binding site residues." needs some qualification. Why is a VDW contact considered "non-specific"? Do the authors imply that buried polar interactions are more stable/rigid than simple non-polar VDW interactions? Complementarity of motion in small molecule ligands is difficult to probe, as noted, but has been examined in protein-protein interactions, particularly in the calmodulin complexes (Marlow et al. Nat Chem Biol 6, 352). Please comment and discuss.

Hydrogen bonds have much stricter distance and angular dependencies compared to van der Waal interactions, making them more stable and rigid. We have updated the text to reflect this thinking, pointing to computational chemistry literature that comments more about the difference between van der Waal interactions and hydrogen bonding in conformational stability in molecular design (lines 528-532).

‘From these results, an intuitive general picture emerges where more specific, directional interactions, such as hydrogen bonds (Bissantz et al. 2010), are more likely to lock the corresponding protein residue in place, thus creating more rigid binding site residues(Majewski et al. 2019). Whereas the more non-specific interactions are correlated with more flexible binding site residues.’

[Editors' note: further revisions were suggested prior to acceptance, as described below.]

The manuscript has been much improved but there are some remaining issues that need to be addressed, as outlined below:The authors explore the effect of different active site definitions on the change in binding site order parameters upon ligand binding. While the authors originally reported this response at a fixed cutoff of 5 Å, they have included new results exploring the change at integer-valued distances from 2-10 Å. This analysis reveals a trend perhaps consistent with expectation, where the median change in order parameter starts high with very tight active site definitions and falls off with increasingly large active sites. In the absence of any surprising features on this curve, the 5 Å cutoff seems reasonable. Please include this figure in the final manuscript (perhaps as a supplemental figure).

We agree that this figure should be included in the supplement. It is now included as Figure 3- Supplementary Figure 2. Additionally, the following is in the text at line 331-333:

We also explored the different binding site cut off values, ranging from 2-10Å observing that the tighter the binding site definition, the more drastic the difference in order parameters between holo and apo pairs (Figure 3- Supplement Figure 2).

The authors evaluated the robustness of the linear regressions between order parameters of non-solvent exposed residues versus average order parameters of binding site residues for holo-apo and apo-apo pairs via bootstrap analysis. F3S2E clearly shows that the mean slopes are robust to exclusion of data in either case. This is a compelling result that greatly improves the paper. Perhaps this result should be emphasized further in the paper.

We agree that this should be included in the paper. We have added the following in the manuscript on line 450-454:

The difference in the slope between the holo-apo and apo-apo dataset was further compared using a bootstrap analysis, demonstrating that the mean slope of the holo-apo is more than two standard deviations away from the apo-apo slope, representing the robustness in differences between the two slopes(p=0.0, z-test; Figure 3—figure supplement 3E).

The authors demonstrated that the majority of the proteins under consideration appear to have an active site composed of a fraction of the total residues less than half of the total number of residues total, and that the number of outliers in this comparison (i.e., with more active site residues than distant residues) is small and thus unlikely to bias the result. The question "how does protein size correlate with the observed effect" is left unanswered and may be scientifically interesting, but perhaps the analysis performed is sufficient to demonstrate that there are at least no pathologies associated with active site definitions. Please comment.

To analyze the impact that protein size has on the relationship between the difference in order parameters in binding site residues versus the residual order parameters in distant residues, we binned proteins based on the number of residues.

We colored each point on our original binding site residues versus the residual order parameters in distant residues based on the protein size and did not observe any clustering. We have included these results in Figure 3- Supplemental Figure 4A.

The following line was added to the manuscript on lines 430-432:

We also explored if protein size impacts our results, but did not observe any trend between protein size and order parameter correlation (Figure 3—figure supplement 3E)

Regarding the comparison to the PKA data from Kim et al. 2017 in point 7, the authors note that a direct comparison is challenging, due to the specifics of the different experiments. The authors clarify that they mean that the overall nature of the changes upon ligand binding are similar (i.e., rigidification throughout the protein). Please clarify the word "pattern" here, as it suggests something more residue-specific, although the overall statement seems fair now.

We believe that the line that discusses the comparison in the patterns observed in PKA and our kinases series adequately states that the trends are similar but not specific residues, as the residue between these proteins are different.

‘This dispersed pattern is similar to the trend of rigidification that is observed by NMR in PKA upon substrate binding, suggesting that changes in conformational dynamics in kinases systems are structurally dispersed as a function of ligand state (Kim et al. 2017)

The addition of F3S2F in point 7 is welcome, although side-by-side comparison with Moorman et al. 2012 F6 is challenging to interpret. Moorman et al. identify a specific subset of contiguous residues in lysozyme which rigidify upon ligand binding, and several that relax. Do the authors intend to imply that the same residues rigidify with ligand binding as in the previous study, or are they saying that there is some more general spatial pattern that is similar? Is there no quantitative comparison that can be made? Explicitly, do the ω-class residues that show collective rigidification upon ligand binding (A11, V29, I55, L83, A90, V92, M105, A107) also rigidify based on the analysis presented here? Do the peripheral residues that relax upon ligand binding (L25, T69) also relax in the analysis presented here? It is very difficult to make such an assessment by staring at F3S2F. There may not be a matching crystal structure for chitotriose, but one might expect some overlap in the residues which rigidify or relax the most for other similar ligands. Please comment and discuss.Please provide some form of quantitative comparison between the Moorman et al. 2012 results and the results here for lysozyme.

To compare our results with Moorman 2012, we looked at the difference in order parameters (holo-apo) from an X-ray lysozyme structure (PDB ID: 4XEN and 4WM2), and from the data deposited with Moorman 2012 (BMRB accession codes 18304 and 18305). We compared the carbon order parameters from Moorman 2012, taking the first chi angle when multiple were measured. Only 4 out of the 8 residues that became completely rigid upon ligand binding in Moorman et al. 2012 [maroon] however the difference in order parameters we observed were much smaller (range: -0.21 to 0.23). Both L25 and T69 [orange] become more flexible upon ligand binding, however, again we see a much smaller difference (-0.09 and -0.113, respectively). Overall, we observed a weak correlation between these two datasets (slope=0.32). These results are summarized in Figure 3- Supplement Figure 4B and Supplementary Table 5.

While we agree it is good to look at general trends, the differences in timescale (NMR is sensitive only to fast motions), ligands, and crystal contacts/solution state make it difficult to compare specific residues from this paper to our investigation here.

The following was added to the manuscript at lines 469-470:

‘Specifically, this difference is greatest between binding residues and non-solvent exposed residues, as previously observed in lysozyme, however there was only weak residue to residue correlation (Figure 3—figure supplement 4A, Figure Supplement 4B) (Moorman, Valentine, and Wand 2012).’

Furthermore, the question about a similar comparison with Moorman et al. 2014 for Ras Cdc42 still needs to be addressed.

Moorman et al. 2014 studied protein-protein interactions (Cdc42Hs and the binding domain of the PAK3 kinase (PDB46)). We did not study nor infer any results or conclusions related to protein-protein interactions. Protein-protein interactions are something we are interested in studying in the future and will examine this case as part of those studies.